

# ChronoLorica – Introduction of a soil-landscape evolution model combined with geochronometers

W.M. van der Meij[1], Arnaud J.A.M. Temme[2], Steven A. Binnie[3], Tony Reimann[1]

[1] Institute of Geography, University of Cologne, Zülpicher Str. 45, 50674 Cologne, Germany
[2] Department of Geography and Geospatial Sciences, Kansas State University, 920 N17th Street, Manhattan, KS, USA
[3] Institute of Geology and Mineralogy, University of Cologne, Zülpicher Str. 49b, 50674 Cologne, Germany

*Correspondence to*: W. Marijn van der Meij (m.vandermeij@uni-koeln.de)

**Abstract**

Understanding long-term soil and landscape evolution can help us understand the threats to current-day soils, landscapes
and their functions. The temporal evolution of soils and landscapes can be studied using geochronometers, such as OSL particle
ages or radionuclide inventories. Also, soil-landscape evolution models (SLEMs) can be used to study the spatial and temporal
evolution of soils and landscapes through numerical modelling of the processes responsible for the evolution. SLEMs and
geochronometers have been combined in the past, but often these couplings focus on a single geochronometer, are designed
for specific idealized landscape positions or do not consider multiple transport processes or post-depositional mixing processes
that can disturb the geochronometers in sedimentary archives.

We present a coupling of soil-landscape evolution model Lorica with a geochronological module, named ChronoLorica.
The module traces spatiotemporal patterns of particle ages, analogous to OSL ages, and radionuclide inventories during the
simulations of soil and landscape evolution. The geochronological module opens rich possibilities for data-based calibration
of simulated model processes, which include natural processes, such as bioturbation and soil creep, as well as anthropogenic
processes, such as tillage. Moreover, ChronoLorica can be applied to transient landscapes that are subject to complex boundary
conditions, such as land use intensification, and processes of post-depositional disturbance which often result in complex geo-
archives.

In this contribution, we illustrate the model functionality and applicability by simulating soil and landscape evolution along
a two-dimensional hillslope. We show how the model simulates the development of two geochronometers: OSL particle ages
and cosmogenic nuclide inventories. The results are compared with field observations from comparable landscapes. We also
discuss the limitations of the model and highlight its potential applications in pedogenical, geomorphological or geological
studies.



## 1        Introduction

Soils and landscapes have been affected by climate change and human use for over thousands of years (Rothacker et al., 2018; Stephens et al., 2019), leading to soil degradation in the form of soil erosion, soil carbon losses and nutrient losses (Sanderman et al., 2017; Olsson et al., 2019). Since the industrial revolution, these degradation processes have been greatly amplified due to deforestation, intensive land management and more extreme weather. Knowledge of the long-term rates and extents of these degradation processes is required to understand the threats to current-day soils, landscapes and their functions.

In eroding and depositionary landscapes, soil material is detached, transported and deposited by natural and anthropogenic processes. These sedimentary deposits provide archives that can be used to derive phases and rates of landscape change, which are essential to understand past and present landscape dynamics (Dotterweich, 2008). There are various methods to study the erosion and sediment dynamics in eroding landscapes. One method is based on soil particles building up an optically stimulated luminescence (OSL) signal that is reset when the particle is exposed to daylight and is recharged by ionizing radiation in the

subsurface when shielded from daylight (Wallinga et al., 2019). This OSL signal acts as a proxy for the duration of burial of the layer in which the soil particle is located. Another method uses radionuclides, which are rare radioactive isotopes that form or accumulate in soils and sediments. The two- or three-dimensional distribution patterns of these radionuclides can be used to calculate soil erosion or deposition rates and even bedrock erosion rates (Ritchie and McHenry, 1990; Brown et al., 1995; Heimsath et al., 1997). Examples of radionuclides are cosmic ray-produced nuclides, such as $^{10}$Be (t1/2 = 1.39 Ma) or $^{14}$C (t$_{1/2}$

= 5.7 ka) (Lal, 1991; Ivy-Ochs and Kober, 2008), and fallout radionuclides, such as $^{137}$Cs (t$_{1/2}$ =30 a) or $^{239+240}$Pu (t$_{1/2}$ = 24.1 ka, 6.55 ka), that are released after nuclear accidents and bomb tests (Ketterer et al., 2004; Alewell et al., 2017). Depending on their half-life, different radionuclides provide information on processes that act over different time scales. Therefore, both OSL signals and radionuclides are the basis for geochronometers, that can be used for studying lateral sediment transport processes, as well as vertical soil mixing processes (e.g., Brown et al., 2003; Arata et al., 2016a, b; Román-Sánchez et al.,

2019b). They are used in experimental studies (Stockmann et al., 2013; Gliganic et al., 2015) as well as numerical studies (Furbish et al., 2018b; Campforts et al., 2016; Johnson et al., 2014).

However, there are settings where geochronometers are not immediately suitable for studying soil and landscape change. This is for example the case in landscapes where sedimentary archives are partially disturbed (e.g., Van der Meij et al., 2019) or absent, or where the sedimentary material is unsuitable for geochronological methods. In other cases, existing

geochronological methods rely on assumptions that are not valid for the landscape studied. For example, most analytical methods for radionuclides assume steady-state conditions in a landscape (Heimsath et al., 1997). This assumption is not always true, certainly not in transient landscapes that are subject to changing erosion rates and anthropogenic disturbances, such as agricultural landscapes (Willenbring and von Blanckenburg, 2010; Hippe et al., 2021).

In these cases, it should be useful to directly simulate the transport and mixing processes of geochronometers using

numerical computer models, specifically soil-landscape evolution models (SLEMs). SLEMs are models that simulate pedogenical and geomorphological processes that are responsible for spatial and temporal soil and landscape development,



driven by various internal and external drivers (Minasny et al., 2015; Van der Meij et al., 2018). By simultaneously considering pedogenical and geomorphological processes, interactions and co-evolution of soils and landscapes can be simulated. SLEMs have the advantage that they provide continuous, landscape-wide soil and landscape properties in three dimensions, as opposed

to field measurements that are typically limited in space. Also, such models provide changes in these properties over time, where measurements often are taken at one point in time, over a very short time-span, or may average rates over excessively long timescales relative to the processes that are studied (Van der Meij, 2022). This four-dimensional representation of the soil-landscape system enables the study of lateral and vertical soil and geomorphological processes, including the development of sedimentary archives.

SLEMs have successfully been combined with radionuclide methods to parametrize and calibrate process rates and increase understanding of soil and landscape change. For example, SPEROS-C and derived models are calibrated with fallout isotopes to study recent anthropogenic erosion processes (Van Oost et al., 2003, 2005; Wilken et al., 2020), Be2D studies the mobility of meteoric $^{10}$Be by simulating vertical and lateral mixing and transport processes (Campforts et al., 2016), and Anderson (2015) and Furbish et al. (2018a, b) developed models (not strictly SLEMs) that study distribution of cosmogenic nuclides and

OSL particle ages due to transport and mixing processes on hillslopes. Individual soil processes can also be calibrated using geochronometers, for example clay translocation (Jagercikova et al., 2015) and bioturbation (Wilkinson and Humphreys, 2005; Johnson et al., 2014; Román-Sánchez et al., 2019b). SLEMs combined with geochronometers, however, often focus on a single geochronometer or soil-landscape process, are designed for specific, idealized landscape positions, and/or do not consider secondary processes that might disturb the geochronometers in sediment archives, such as post-depositional mixing by tillage.

In this study, we aim to bridge these limitations by combining SLEMs with geochronometers. We present an extension to the SLEM Lorica (Temme and Vanwalleghem, 2016; Van der Meij et al., 2020), where we coupled various geochronometers to the existing pedogenical and geomorphological processes in the model. With this extended model, named ChronoLorica, we aim to:

1. Calibrate pedogenical and geomorphological processes in the model using measured geochronometers;

2. Study the effect of natural and anthropogenic soil transport and mixing processes on the chronological information present in soils and sediment archives;

3. Understand soil and landscape evolution in transient agricultural landscapes that are subject to complex boundary conditions, such as intensification of land management.

In this paper, we introduce ChronoLorica and show its potential for unravelling chronologies and landscape evolution in

transient landscapes. First, we describe the geochronological module of ChronoLorica. Second, we simulate several soil and landscape processes to show how chronologies can develop in natural and transient agricultural landscapes. Third, we discuss the added value of coupling soil-landscape evolution modelling with geochronometers, the model limitations and potential applications of the model in soil, geomorphological or geological studies.



## 2 Model description

### 2.1 Model architecture

ChronoLorica is an extension to soil-landscape evolution model Lorica (Temme and Vanwalleghem, 2016). In Lorica, the relief of a landscape is represented by a raster-based digital elevation model (DEM), which controls the overland routing of water. Elevation of each raster cell can be modified by removal or addition of soil material by various geomorphological and pedogenical processes. Soils are represented by a pre-defined number of soil layers below each raster cell. Lorica works with 100 a dynamic adaptation of the number and thickness of soil layers, enabling more detail in heterogeneous sections and less detail in homogeneous sections of the soil. Inside these layers, the model keeps track of five texture classes (coarse, sand, silt, clay, fine clay) and two organic matter (OM) classes (old and young). The composition of each layer can be modified by geomorphological processes as well as pedogenical processes. The soil components are recorded in kilograms and converted to elevation or layer thickness in meters using a pedotransfer function of bulk density (here, we use Tranter et al., 2007). The 105 dynamic architecture of Lorica makes it suitable for adaptation to locally occurring processes (Van der Meij et al., 2016) or including additional drivers (HydroLorica, Van der Meij et al., 2020). Lorica and ChronoLorica are written in C#.

### 2.2 Process descriptions

The pedogenical processes that are currently included in the model are physical and chemical weathering, clay translocation, bioturbation and carbon cycling. The pedogenical processes are vertically oriented, meaning that transport due to these 110 processes occurs only in the vertical direction. The geomorphological processes that are currently included are water erosion and deposition by overland flow, tillage mixing and erosion, soil creep and tree throw. The geomorphological processes are oriented laterally, leading to detachment, transport and possible deposition of soil material. The mixing processes associated with these processes occur vertically.

These processes can be considered as advective processes, diffusive processes or transformative processes. Advective 115 processes trigger directional movements of soil matter, often driven by water flow, leading to heterogeneity in topography or development of soil horizons. Diffusive processes are processes that homogenize soil layers or topography, for example by mixing. Transformative processes transform soil particles into other particles, for example by breaking down due to weathering. The mathematical descriptions of these processes follow advective and diffusive formulations that are often used in numerical particle transport models (e.g., Anderson, 2015; Furbish et al., 2018a), with the difference that the processes in 120 Lorica are programmed as distinct processes that affect multiple soil and landscape properties, including geochronometers.

In this paper, we simulated the processes of soil creep, tillage, clay translocation and bioturbation (See Section 0). These processes are described in detail below. For descriptions of the other processes in the model, we refer to earlier publications (Temme and Vanwalleghem, 2016; Van der Meij et al., 2020).

Soil creep is simulated as a diffusive geomorphological process, that leads to gradual smoothing of the surface. Soil material 125 is transported between soil layers of adjacent cells. The amount of soil material creeping out of a cell towards a lower lying



neighbouring cell, $CR_{local}$ [kg a$^{-1}$], is calculated by multiplying the potential amount of creep $CR_{pot}$ [kg a$^{-1}$] with an exponential depth decay function over the soil depth $sd$ [m], the local slope $\Lambda_{local}$ [%] and the division of the slope gradient $\Lambda$ [%] to the power of factor $p$ [-] towards the neighbouring cell, divided by the sum of slope gradients to the power of $p$ towards all lower-lying neighbouring cells (Eq. 1). This equation differs from earlier reported process descriptions in Lorica (Van der Meij et al., 2020). The shape of the depth decay function is controlled by the depth decay rate for creep $dd_{CR}$ [m$^{-1}$]. $CR_{local}$ is divided over all soil layers at the source location, proportionally to the fraction of the integral of the depth decay function over the upper and lower depths ($z_{upper}$, $z_{lower}$) [m] of the respective layer, divided by the integral of the depth decay function over the entire soil column (Eq. 2). The resulting $CR_{layer}$ [kg a$^{-1}$] is the total amount of mass leaving a soil layer. This mass is gathered from all fine texture fractions, relative to the texture distribution. The mass is transported to adjacent soil layers in the receiving cell. When multiple receiving layers neighbour the source layer, $CR_{layer}$ is distributed proportional to the size of the shared boundaries.

$$CR_{local} = CR_{pot} * (1 - e^{-dd_{CR}*sd}) * \frac{\Lambda_{local}^p}{\sum_j^J \Lambda_j^p} * \Lambda_{local} \tag{1}$$

$$CR_{layer} = CR_{local} * \frac{\int_{z_{upper}}^{z_{lower}}(e^{-dd_{CR}*z})}{\int_0^{sd}(e^{-dd_{CR}*z})} \tag{2}$$

Tillage consists of two parts: homogenization of the topsoil and transport of soil material. All soil layers that are located in the range of the plough depth $pd$ [m] are completely mixed with each other in every timestep, where layers that are partially in the plough layer contribute a fraction to the mixture. Local tillage transport $TI_{local}$ [m] is calculated in a similar way as creep, by multiplying a tillage constant $C_{til}$ [a$^{-1}$] with the slope gradient $\Lambda_{local}$ [m m$^{-1}$] to the power of a convergence factor $p$ [-] divided by the sum of all slope gradients to the power $p$ and the plough depth $pd$ (Eq. 3). $TI_{local}$ is calculated in meters per year. The ratio between $TI_{local}$ and the thickness of each possibly eroding soil layer is used to determine the fraction of soil material that can be eroded out of that layer. The eroded material is usually taken from the topmost layer, but can be taken from subsequent layers as well, when the eroded thickness exceeds the thickness of the top layer.

$$TI_{local} = C_{til} \frac{\Lambda_{local}^p}{\sum_j^J \Lambda_j^p} * \Lambda_{local} * pd \tag{3}$$

Clay translocation is calculated using an advection equation, similar to Jagercikova et al., (2015). Bioturbation (see next paragraph) serves as the diffusive part of the process, where clay particles are bioturbated between soil layers. The translocation of clay from a certain soil layer, $CT_{layer}$ [kg a$^{-1}$], is calculated by multiplying the advection parameter $CT_{adv}$ [m a$^{-1}$] with a depth decay function, containing depth decay parameter $dd_{CT}$ [m$^{-1}$] and depth of the layer $z_{layer}$ [m], the clay fraction of that layer $fclay_{layer}$ [-], the bulk density of that layer $BD_{layer}$ [kg m$^{-3}$] and the cell area $dx^2$ [m$^2$] (Eq. 4). The translocated clay is transported to the underlying layer, or, in the case of the lowest layer, is lost from the soil column. Not all clay in the soil is available for transport. We used the equation of Brubaker et al., (1992) to estimate the fraction of clay that is water-dispersible, i.e. available for translocation ($fclay_{wd}$, Eq. 5). We estimated the CEC of each layer, $CEC_{layer}$, which is required for the equation of Brubaker,





using a pedotransfer function from Ellis and Foth (1996), using the clay fraction $fclay_{layer}$ and organic matter fraction $fOM_{layer}$ in the respective layer (Eq. 6). The latter is 0 in our simulations, because we didn't simulate the SOM cycle.

$$CT_{layer} = CT_{adv} * e^{-dd_{CT}*z_{layer}} * fclay_{layer} * BD_{layer} * dx^2 \tag{4}$$

$$fclay_{wd} = 0.01 * \left(0.369 * fclay_{layer} * 100 - 8.96 * \frac{CEC_{layer}}{fclay_{layer}*100} + 4.48\right) \tag{5}$$

160
$$CEC_{layer} = 0.1 * \left(32 + 3670 * \frac{fOM_{layer}}{1.72} + 196 * fclay_{layer}\right) - 300 * \frac{fOM_{layer}}{1.72} \tag{6}$$

Bioturbation is calculated as a diffusive process mixing the soil. The total amount of bioturbation at a certain location $BT_{local}$ [kg a$^{-1}$] is calculated by multiplying the potential bioturbation $BT_{pot}$ with a depth decay function that contains the depth decay parameter for bioturbation $dd_{BT}$ and the local soil depth $sd$ (Eq. 7). The local amount of bioturbation is divided over all soil layers at that location, proportional to the fraction of the integral of the depth decay function over the upper and lower depths 165 ($z_{upper}$, $z_{lower}$) of the respective layer, divided by the integral of the depth decay function over the entire soil column, similar to the creep calculations (Eq. 8). The resulting layer bioturbation, $BT_{layer}$ [kg a$^{-1}$], is the amount of mass that is exchanged with the respective layer. The exchange occurs with all other layers, but the amount of exchange decreases exponentially with distance to the source layer. For this exponential function, we use a depth decay constant of two times the $dd_{BT}$ to limit the distance over which soil material is mixed. $BT_{layer}$ is derived from all fine soil textures and organic matter classes, proportional 170 to their fractions in the soil.

$$BT_{local} = BT_{pot} * (1 - e^{-dd_{BT}*sd}) \tag{7}$$

$$BT_{layer} = BT_{local} * \frac{\int_{z_{upper}}^{z_{lower}}(e^{-dd_{BT}*z})}{\int_0^{sd}(e^{-dd_{BT}*z})} \tag{8}$$

## 2.3    Geochronometers

We built in two types of geochronometers in Lorica. These are the burial ages of individual soil particles, analogous to OSL 175 ages, and the inventories of various cosmogenic nuclides.

### 2.3.1    OSL particle ages

OSL dating determines the last exposure of a soil particle to daylight, i.e. the moment of bleaching. This is helpful for determining the time of burial of a sediment layer, or for determining the rate at which surface particles are mixed in the subsoil. The OSL age is determined by dividing the radiation dose received by a subsample of soil or sediment particle(s) 180 (termed palaeodose, [Gy]) by the ionizing radiation from the surrounding soil, sediments and cosmic rays (termed dose rate, [Gy ka$^{-1}$]). OSL ages can be determined for bulk sample material or for smaller amounts of material down to even single grains of sand (Duller, 2008).

Chronolorica's OSL particle age module keeps track of the erosion, transport and bleaching of a small number of particles in each soil layer through time. Because of this, the intermediate steps of determining palaeodoses and dose rates can be




skipped and the ages of individual particles are traced directly. By tracing the ages of individual particles, we are able to derive
their age proxies, which can be translated into probability density functions (PDFs) of OSL particle ages for each soil layer in
the model. These PDFs can be compared with measured OSL age PDFs for calibration and validation purposes.

The number of particles in each soil layer varies over space and time due to transport and mixing processes. The
redistribution of the particles currently follows the redistribution of the sand fraction in the model, which is the texture class
that that is typically targeted for single-grain OSL dating (Duller, 2008). The sand content of the soil layers in the model is
redistributed due to transport and mixing processes. Because the number of particles that is traced is much lower than the
number of particles represented by the total amount of sand typically present in soil layers, we need to work with probabilities
for particle transport. The probability that a certain particle is transported together with the sand, $P_{transport}$, is equal to the fraction
of the total amount of sand that is transported out of that layer (Eq. 9). In case an entire layer is eroded, all particles are eroded.
In case only 0.1% of the sand is transported, there is a probability of 0.1% for each particle that it is transported as well.

$$P_{transport} = \frac{sand\ transported\ [kg]}{total\ sand\ present\ [kg]} \tag{9}$$

The age of the particles in the model can be set to zero, i.e. the particles can be bleached, when they are located in the top
soil layer. Estimates of the depth that daylight can penetrate the soil to bleach particles range between 1 and 10 mm (Furbish
et al., 2018b), which agrees with long-term bleaching depths in rock surfaces (10 mm, Sellwood et al., 2019). With intensive
mixing of the topsoil, the bleached particles can be found in abundance over the top 5 to 10 cm of the soil (Wilkinson and
Humphreys, 2005). The thickness of the top soil layer is set to the bleaching depth, resulting in a completely bleached upper
soil layer. The initial number of particles per layer depends on the sand content of the layer, and is provided in particles per kg
m$^{-2}$ sand. This is necessary to account for varying bulk density with depth, which also changes the sand contents and the
transport probabilities of the layers. As an example, a soil layer of 0.05 m with 25% sand, bulk density of 1500 kg m$^{-3}$ and 4
simulated particles per kg m$^{-2}$ sand will have 0.05 * 1500 * 0.25 * 4 = 75 particles.

ChronoLorica traces three types of ages or particle properties that are useful in soil mixing and erosion studies. The first
age is the *apparent age*, which is the actual age of a particle that is analogous to an OSL-measured age. This is the age of the
particle that is reset when it is bleached at the surface. Next to that, we also track the *depositional age*, which is reset when a
particle is transported laterally due to surface erosion processes, such as water erosion and tillage erosion. Especially in
agricultural systems, tillage is a strong secondary mixing process that can greatly disturb the depositional chronology in
colluvial deposits, by resurfacing and bleaching particles that may have been deposited a long time ago (Van der Meij et al.,
2019). By tracing both the apparent ages and the deposition ages, we can reconstruct depositional chronologies in soil-
landscape systems that are significantly impacted by post-depositional mixing. The third property that is traced, the *surfaced*
count, is the number of times a particle has been bleached at the surface.

To illustrate the different processes that affect the particle location and age, we will follow the fate of a hypothetical sand
particle in Weichselian (i.e. late Pleistocene) cover sand. This particle has been deposited near the surface, high up on a gently
sloping hillslope. After the landscape stabilized and vegetation started to grow in the Holocene, sands started to get mixed





vertically by bioturbation. Our particle reached the surface several times, where it was bleached, before it was mixed back into the subsurface. The apparent age of the particle corresponds to the last moment the particle was exposed to daylight, while the
surfaced count keeps track of the number of times the particle has been surfaced. At a certain moment in time, the hillslope is cultivated and the soils begin to be ploughed. Our particle is located near the surface at that time, so it is incorporated in the plough layer, which has much higher mixing rates than the undisturbed natural soils. In the plough layer, our particle is surfaced several more times, and its apparent age gets reset every time. Next to the intensive mixing, the particle is also transported downslope due to tillage erosion. Every time the particle moves laterally with the eroding topsoil, the deposition age gets reset.
Eventually, the particle reaches the foot slope, where it is incorporated in a colluvial layer. The deposition age starts to build up, but the apparent age still gets reset when the particle is surfaced in the plough layer at the location of deposition. Only when the particle is buried below new colluvium, and it thus left the active mixing zone, can its apparent age increase again. This is the age that can be measured with OSL dating of the colluvium.

Because the particles are transported between multiple soil layers and locations, the number of particles per layer is not
constant. With erosion, the number of particles in a layer can decrease, while deposition can increase the number of particles. The tracing of individual particles that vary in number for each location required a memory-intensive implementation in the model. We make use of a three-dimensional jagged array for this purpose. Jagged arrays contain elements that can be variable in length. In practice, this means that for every row, column and layer in the model, there is an array of OSL particle ages, that can vary in length. Because of memory restrictions and computational demands, the number of particles in each layer is limited.
This number should depend on the dimensions of the soil landscape that is simulated, the runtime of the model and the specifications of the computer. We will discuss the choice of the initial number of particles in more detail in the discussion.

### 2.3.2    Cosmogenic nuclides

ChronoLorica records the inventories of various cosmogenic nuclides for each row, column and soil layer in the model. This enables the calculation of spatial distribution patterns as well as depth functions of these radionuclides. For each soil
layer, the total number of radionuclides is stored as atoms per $cm^{-2}$. In this contribution, we focus on cosmogenic nuclides. We distinguish externally produced, meteoric cosmogenic nuclides and in-situ produced cosmogenic nuclides. Each radionuclide in the model behaves similarly, but their dynamics depend on the type of accumulation or production, their decay rate and the soil texture fraction they are associated to. It is likely that there are already radionuclides present at the start of a simulation. Therefore, all radionuclides can have an inherited inventory, which is the number of atoms present at the start of simulations.
These inventories are homogeneous throughout the profile.

**Meteoric cosmogenic nuclides**

Meteoric cosmogenic nuclides are produced in the atmosphere via spallation reactions by collision of cosmic radiation with atmospheric gases. These nuclides are then delivered to the earth's surface by capture with precipitation (Willenbring and von Blanckenburg, 2010). For ChronoLorica, we use meteoric $^{10}$Be as one of the meteoric geochronometers. In non-acidic soils,



meteoric [10]Be binds primarily to clay particles (up to 80%, Jagercikova et al., 2015). In more acidic soils (pH < 4.1, Graly et al., 2010; Willenbring and von Blanckenburg, 2010), meteoric [10]Be leaches from the soil.

As Lorica is mainly designed for pedogenesis in loamy soils, meteoric [10]Be is a suitable marker for lateral and vertical clay redistribution, due to erosion processes, bioturbation and clay translocation (Campforts et al., 2016; Jagercikova et al., 2015). The adsorption of meteoric [10]Be is grain-size dependent, but most of the nuclides adsorb to the clay fraction (Wittmann et al.,

2012). To account for this grain-size selectivity, we assign 80% of the input to be associated with the mobile clay fraction (Jagercikova et al., 2015), while the rest is associated to the silt fraction. Vertical and lateral redistribution of meteoric [10]Be in the model thus follows the redistribution of primarily the clay and secondarily the silt fraction.

In ChronoLorica, the deposition of meteoric radionuclides follows an exponentially declining rate with soil depth (Willenbring and von Blanckenburg, 2010). The total local input of a meteoric radionuclide, $A_{me,local}$ [atoms cm$^{-2}$ a$^{-1}$], is

calculated by multiplying the potential input, $A_{me,pot}$ [atoms cm$^{-2}$ a$^{-1}$], which is a given input parameter, with a depth decay function, that contains the adsorption coefficient for the meteoric cosmogenic nuclide, $k_{me}$ [m$^{-1}$], and soil depth $sd$ [m] (Eq. 10). The adsorption coefficient serves the same purpose as the depth decay parameters for the soil processes. $A_{me,local}$ is divided over all soil layers at that location, proportionally to the fraction of the integral of the depth decay function over the upper and lower depths ($z_{upper}$, $z_{lower}$) of the respective layer, divided by the integral of the depth decay function over the entire soil column

(Eq. 11). The change in radionuclide inventory in a certain layer is the sum of the current inventory and $A_{me,layer}$, multiplied with $1-\lambda_{me}$, to account for radioactive decay (Eq. 12).

$$A_{me,local} = A_{me,pot} * (1 - e^{-k_{me}*sd}) \tag{10}$$

$$A_{me,layer} = A_{me,local} * \frac{\int_{z_{upper}}^{z_{lower}}(e^{-k_{me}*z})}{\int_0^{sd}(e^{-k_{me}*z})} \tag{11}$$

$$C_{me,layer,t+1} = (C_{me,layer,t} + A_{me,layer}) * (1 - \lambda_{me}) \tag{12}$$

**In-situ cosmogenic nuclides**

In contrast to meteoric cosmogenic nuclides, in-situ cosmogenic nuclides are produced in the soil, or in the underlying bedrock, itself. Two examples of in-situ cosmogenic nuclides are in-situ [10]Be and in-situ [14]C. Due to the long half-life of [10]Be, this isotope is suitable to trace both short- and long-term soil-landscape processes ($10^2$-$10^7$ a). The shorter half-life of [14]C makes it more suitable to trace Holocene soil-landscape processes ($10^2$-$10^4$ a) (Walker, 2005). Within the uppermost few

meters from the surface, most in-situ cosmogenic nuclides are formed due to penetrating cosmic radiation rays causing nuclear spallation of target elements, such as O in quartz, the mineral most commonly used for in-situ [10]Be and [14]C. Formation of in-situ cosmogenic nuclides can follow other production pathways as well, for example via negative muon capture or neutron reactions induced by fast muons (Dunai, 2010). The contribution of muon reactions to the total production in the upper meters of the earth's surface ranges from a few percent for in-situ [10]Be to over 20% for in-situ [14]C (Balco, 2017; Lupker et al., 2015).

Production from muons is however still poorly understood and quantified, which making this a potential source of uncertainty, especially for in-situ [14]C, which has a substantial muogenic production pathway (Balco, 2017; Hippe, 2017). That is why we only use in-situ [10]Be in this study.





The production of in-situ cosmogenic nuclides via spallation and muogenic reactions is described as two, or more, exponential functions (Lal, 1991; Braucher et al., 2013). Although describing the complex muogenic production of in-situ

cosmogenic nuclides as a single exponential function is a simplification, it's accurate enough for most geological applications (Balco, 2017), and fits with the reduced complexity of the Lorica model.

The annual change in the inventory of an in-situ cosmogenic nuclide $C_{is}$ [#atoms cm$^{-2}$] is depth-dependent and follows Eq. 13-15 (Lal, 1991; Braucher et al., 2013):

$$C(z)_{is,t+1} = \left( C(z)_{is,t} + (P(z)_{is,sp} + P(z)_{is,mu}) * \frac{sand\ content(z)}{(dx*100)^2} \right) * (1 - \lambda_{is}) \tag{13}$$

With

$$P(z)_{is,sp} = P(0)_{is,sp} * \exp\left(\frac{-z\rho}{\Lambda_{sp}}\right) \tag{14}$$

$$P(z)_{is,mu} = P(0)_{is,mu} * \exp\left(\frac{-z\rho}{\Lambda_{mu}}\right) \tag{15}$$

Here, $P(0)_{is}$ [atoms g quartz$^{-1}$ a$^{-1}$] is the annual production rate of the radionuclide at the soil surface via spallation ($sp$) or muogenic ($mu$) reactions, $z$ is the depth below the surface [m], P(z)$_{is}$ is the production rate of the in-situ cosmogenic nuclide

at depth $z$, $\rho$ is the average bulk density of the material overlying the layer at depth $z$ [kg m$^{-3}$] and $\Lambda_{sp}$ and $\Lambda_{mu}$ are the attenuation lengths for spallation and muogenic production [kg m$^{-2}$]. $\lambda_{is}$ is the decay rate of the cosmogenic nuclide, *sand content(z)* [kg] is the mass of the sand fraction at depth $z$ and *dx* [m] is the cell size. *sand content(z)* and *dx* are used to recalculate the production in atoms g quartz$^{-1}$ a$^{-1}$ to atoms cm$^{-2}$ a$^{-1}$. For simplicity, we assume that the quartz fraction of the sand content in the model is constant, and that therefore the sand content [kg] in each layer determines how many atoms are produced.

Mohren et al. (2020) and Evans et al. (2021) point out the sensitivity of this method to estimates of the bulk density of the soil, which is often assumed constant. ChronoLorica calculates spatially explicit bulk densities based on soil texture, organic matter properties and soil depth using a pedotransfer function (here we use Tranter et al., 2007), which helps accounting for variations in bulk density.

In-situ cosmogenic $^{10}$Be is formed in and most often measured from quartz particles. Therefore, we linked the lateral and

vertical redistribution of the in-situ cosmogenic nuclides to the sand fraction in the model. Because the redistribution of all in-situ cosmogenic nuclides follows the redistribution of the sand fraction, their redistribution patterns will be similar.

## 3    Experimental set-up

To illustrate and test the behaviour and functionalities of ChronoLorica, we simulated a variety of processes along an artificial one-dimensional hillslope (Figure 1).





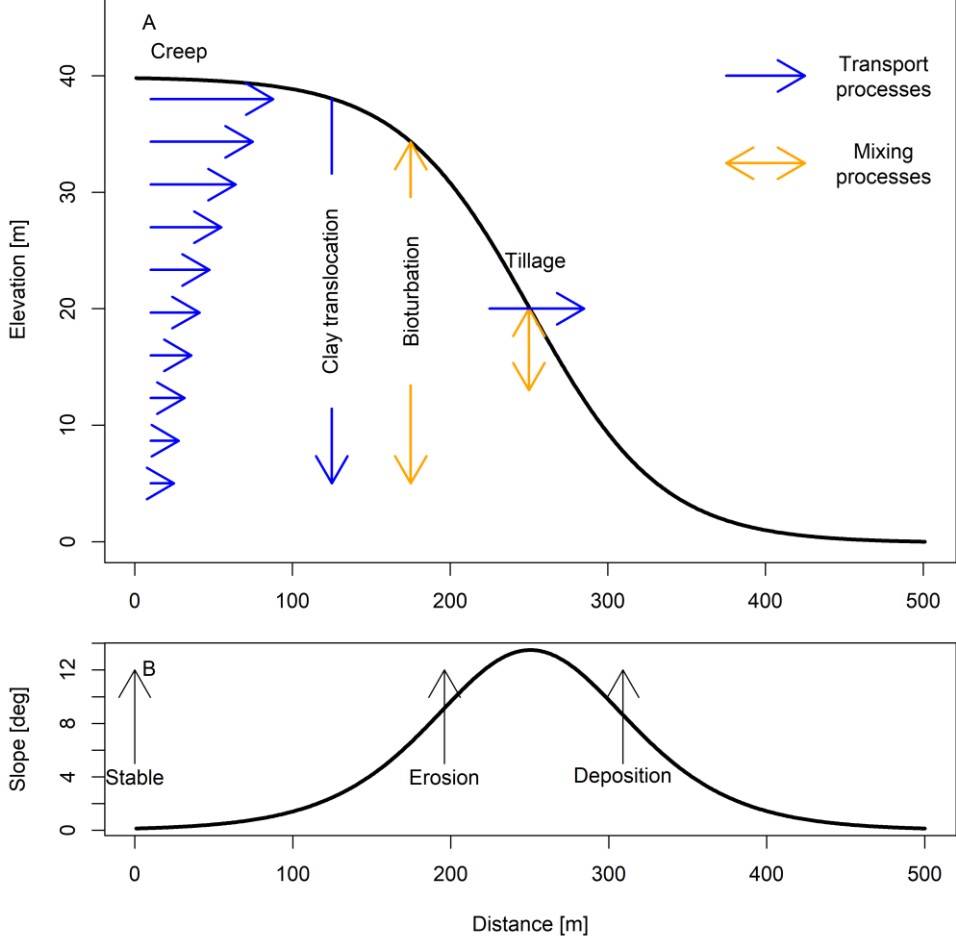

**Figure 1: A: Elevation transect of the input hillslope for the simulations with ChronoLorica. The arrows show which processes were simulated and how they affect soil particles and chronological tracers. B: The corresponding slope transect of the input elevation. The arrows indicate the locations of the different landscape positions that are used in the data presentation.**

The hillslope has the shape of a gaussian curve, containing stable, eroding and depositional hillslope positions. The parent

material of the soils was set based on loess sediments, with 25% sand, 60% silt and 15% clay. The initial soil thickness was

set at 3 m, divided over 50 soil layers with a thickness of 0.05 m, with the top layer having a thickness of 5 mm, representing

the OSL bleaching depth, and lowest layer receiving the remainder of the soil thickness. The lowest soil layer contained the

residual soil depth. The simulations started with 10 ka of natural soil, landscape and chronology development, where we

simulated the processes of creep, bioturbation and clay translocation. This natural period was followed by 500 years of

agricultural land use, by introducing mixing and erosion by tillage. Figure 1 shows how the different processes affect transport

and mixing of soil particles and chronological tracers. We will show how particles, in-situ [10]Be and meteoric [10]Be are





redistributed due to the different processes, because these chronological tracers are commonly used in erosion and soil mixing studies.

The goal of this simulation was to provide insight in the functioning and applicability of the model, rather than trying to reproduce measured chronologies. The selected processes, parameters, input hillslope and periods of land use are therefore a simplification of real-world development of landscapes, but this simplification suffices to illustrate the aims of ChronoLorica. The model parameters require constraining with experimental data when the model is applied to real-world settings.

Table 1 shows the parameters used in this study. Where possible, the parameters were based on field evidence. Where that wasn't possible, the parameters were estimated based on previous model studies to get illustrative outcomes.


**Table 1: Parameters used for the geochronometers, pedological and geomorphological processes in ChronoLorica for this study. When the reference states 'est', the parameter is estimated.**

| | Marker/process | Parameter | Symbol | Value | Reference |
|---|---|---|---|---|---|
| **Particle ages** | OSL Particle ages | Initial number of particles per layer [particles kg sand$^{-1}$ m$^{-2}$] | | 4 | - |
| | | Bleaching depth [m] | | 0.005 | (Furbish et al., 2018b) |
| **Ex-situ radionuclides** | Meteoric $^{10}$Be | Annual input [atoms cm$^{-2}$ a$^{-1}$] | $A_{Be-10,met,pot}$ | $1*10^6$ | (Willenbring and von Blanckenburg, 2010) |
| | | Decay constant [a$^{-1}$] | $\lambda_{Be-10}$ | $4.99*10^{-7}$ | (Chmeleff et al., 2010; Korschinek et al., 2010) |
| | | Adsorption coefficient | $K_{Be-10}$ | 4 | (Willenbring and von Blanckenburg, 2010) |
| | | Inherited inventory [atoms g soil$^{-1}$] | $C_{Be-10,met,t=0}$ | $0.2*10^8$ | (Calitri et al., 2019) |
| | | Clay-associated fraction | | 0.8 | (Jagercikova et al., 2015) |
| **In-situ radionuclides** | General in-situ CNs | Attenuation length spallation production [kg m$^{-2}$] | $\Lambda_{sp}$ | 1600 | (Gosse and Phillips, 2001) |
| | | Attenuation length muogenic production [kg m$^{-2}$] | $\Lambda_{mu}$ | 25000 | (Balco, 2017) |
| | In-situ $^{10}$Be | Spallation production rate at the surface assuming LSD$_n$ scaling [atoms g quartz$^{-1}$ a$^{-1}$] | $P(0)_{(Be-10,is,sp)}$ | 3.92 | (Borchers et al., 2016) |
| | | Muogenic production rate at the surface [atoms g quartz$^{-1}$ a$^{-1}$] | $P(0)_{(Be-10,is,mu)}$ | 0.084 | (Balco, 2017) |
| | | Inherited inventory [atoms g quartz$^{-1}$] | $C_{Be-10,is,t=0}$ | 65000 | (Calitri et al., 2019) |



| | | | | | |
|---|---|---|---|---|---|
| Pedogenical processes | Bioturbation | Potential bioturbation rate [kg m$^{-2}$ a$^{-1}$] | $BT_{pot}$ | 10 | Est |
| | | Depth decay rate [m$^{-1}$] | $dd_{BT}$ | 5 | Est |
| | Clay translocation | Surface advection [m a$^{-1}$] | $CT_{adv}$ | 0.0025 | Est |
| | | Depth decay rate [m$^{-1}$] | $dd_{CT}$ | 10 | Est |
| Geomorphological processes | Soil creep | Potential creep rate [kg m$^{-2}$ a$^{-1}$] | $CR_{pot}$ | 10 | Est |
| | | Depth decay rate [m$^{-1}$] | $dd_{CR}$ | 5 | Est |
| | Tillage | Tillage constant [a$^{-1}$] | $C_{til}$ | 2 | Est |
| | | Ploughing depth [m] | $pd$ | 0.25 | (Van der Meij et al., 2019) |
| | | Convergence factor [-] | $p$ | 2 | (Temme and Vanwalleghem, 2016) |
| Initial soil properties | Initial soil properties | Sand fraction [-] | | 0.25 | - |
| | | Silt fraction [-] | | 0.6 | - |
| | | Clay fraction [-] | | 0.15 | - |
| | | Initial thickness [m] | | 3 | - |
| | Layer properties | Initial layer thickness [m] | | 0.05 | - |
| | | Number of layers | | 50 | - |



# 4      Results



**Figure 2: Evolution of chronologies at a stable position in the simulated landscape in the first 10ka of simulations (natural phase). The shading of the lines indicates the time in the model simulations. A: Evolution of the average OSL particle age-depth plot over time. B: OSL age distributions of particles in a selection of soil layers after 10ka of simulations. C: Evolution of the meteoric $^{10}$Be-depth plot over time. The clay-depth profile at time step 10000 is indicated in blue. D: Evolution of the in-situ $^{10}$Be-depth plot over time.**

Figure 2 shows the evolution of different chronological markers for the natural phase of soil and landscape evolution. In this phase, no anthropogenic processes were simulated. The changes in the chronologies can be attributed to the natural processes of soil creep, bioturbation and clay translocation.




The OSL particle age-depth plot (Figure 2A) shows an increasingly steep depth profile. At the bottom of the profile, below
1.5 meters, the average OSL particle age increases equally with the simulation time. Closer to the surface, the average OSL particle age is increasingly deviating from the simulation time throughout the simulations. In the top soil layer, which is completely bleached, the particles have an average age of zero years. Over time, the depth at which the rejuvenated ages can be found increases. The wiggles in the age-depth curves are due to the limited number of particles per layer. With an increasing number of particles, the curves become smoother. Figure 2B shows the age distributions after 10ka of simulations. In the
subsurface, the majority of the particles have an age of 10ka, equal to the simulation time, while layers closer to the surface contain an increasing number of particles with younger ages, which were mixed into the subsoil due to bioturbation. The age of these rejuvenated particles increases with depth, as it takes time to mix these particles deeper into the soil. At a depth of 1.5 meters, almost no younger particles are present.

The meteoric $^{10}$Be-depth profile (Figure 2C) develops a bulge shape, typical for meteoric $^{10}$Be in soil profiles (Graly et al.,
2010). The shape of the bulge closely follows the shape of the clay-depth profile, due to the partial association of meteoric $^{10}$Be to the clay fraction. Effects of bioturbation are not clearly visible in the profiles. The profiles show continuously increasing inventories over the entire profile, indicating the inventories are not yet in equilibrium, which was also not expected over the simulated time scales. Below 1.5 meters, the inventories have a constant value of $2*10^7$ atoms g soil$^{-1}$, which was the inherited inventory for meteoric $^{10}$Be.

The in-situ $^{10}$Be-depth profile (Figure 2D) also shows continuously increasing inventories and a disequilibrium over time. The lower parts of the profiles follow an exponential curve, but the upper part of the curve, above 0.75 m, deviates from that curve. Here, the inventories become more similar towards the soil surface, showing homogenization effects of bioturbation. The inventories of in-situ $^{10}$Be increase over the entire profile throughout the simulations.

Figure 3 shows the changes in chronological markers at different landscape positions, due to the simulation of tillage erosion
in the agricultural phase.

At the relatively stable position, the elevation of the soil surface shows a small decrease of 0.10 meters. In the plough layer, the OSL particle ages show a large decrease compared to the age-depth profile in the natural phase, with average OSL particle ages around 0.5 ka. Most of the particles in the plough layer have been bleached due to the intensive mixing, but the unbleached particles have a disproportionate effect on the average ages. This effect is visualized in Figure 4, which shows more detailed
age information for the depositional location. Below the plough layer, there is also a reduction in the average age. This is a consequence of particles bleached in the plough layer, that are mixed deeper in the profile due to bioturbation.

At the stable position, the meteoric $^{10}$Be shows a homogenized inventory in the plough layer (Figure 3). The inventories are not completely homogeneous, because of the unit the inventories are expressed in. Deviations in the soil mass cause the small deviations in the concentrations of meteoric $^{10}$Be. When expressed in atoms/g silt and clay, the fraction that the meteoric $^{10}$Be
is associated to, the inventories are identical for every layer in the plough layer. For the in-situ $^{10}$Be, expressed in atoms/g sand, there is also a homogeneous inventory in the plough layer. These chronological markers clearly show the homogenizing effect of the intensive mixing in the plough layer.



**Figure 3: Changes in the simulated chronologies due to tillage erosion in three different landscape positions (for locations, see Figure 1). The Y-axes show absolute elevation of the soil layers, to illustrate elevation changes due to erosion and deposition. The black lines indicate the end of the natural phase and the red lines the end of the agricultural phase. The red band indicates the depth of the plough layer at the end of the agricultural phase.**

The erosion position shows a decrease in elevation of 1.04 meters. This leads to a truncation of the cosmogenic nuclide depth-profiles. Aside from the mixing in the plough layer, the depth profiles follow the same trend as the natural depth-profiles. In the plough layer, the inventories are higher compared to the natural profile at the same depth, due to incorporation of higher inventories from higher in the profile. The age-depth profiles also show a truncation, where the subsurface profiles are similar. In the plough layer, the OSL particle ages are again much younger compared to the natural setting, but older compared to the





particles in the plough layers at the stable and deposition location. Due to the erosion, unbleached particles from the subsurface are incorporated in the plough layer, which increase the average OSL particle age.

The deposition location has an elevation increase of 1.04 meters. At this location, the effects of tillage are two-fold. First, the chronological markers were disturbed in the plough layer, homogenizing the cosmogenic nuclides and resetting OSL particle ages, similar to the stable position. Second, colluvial material was transported towards this location, building up a layer of colluvium, with additional cosmogenic nuclides and particles. The average OSL particle ages in the plough layer are similar to those in the stable position. In the colluvial profile below the plough layer, OSL particle ages slowly increase up to

the depth where the soil hasn't been affected by tillage. Meteoric [10]Be shows a new bulge shape developed in the colluvial profile. This is due renewed clay translocation in the colluvium. The in-situ [10]Be shows a slowly decreasing inventory in the colluvial profile towards the surface.

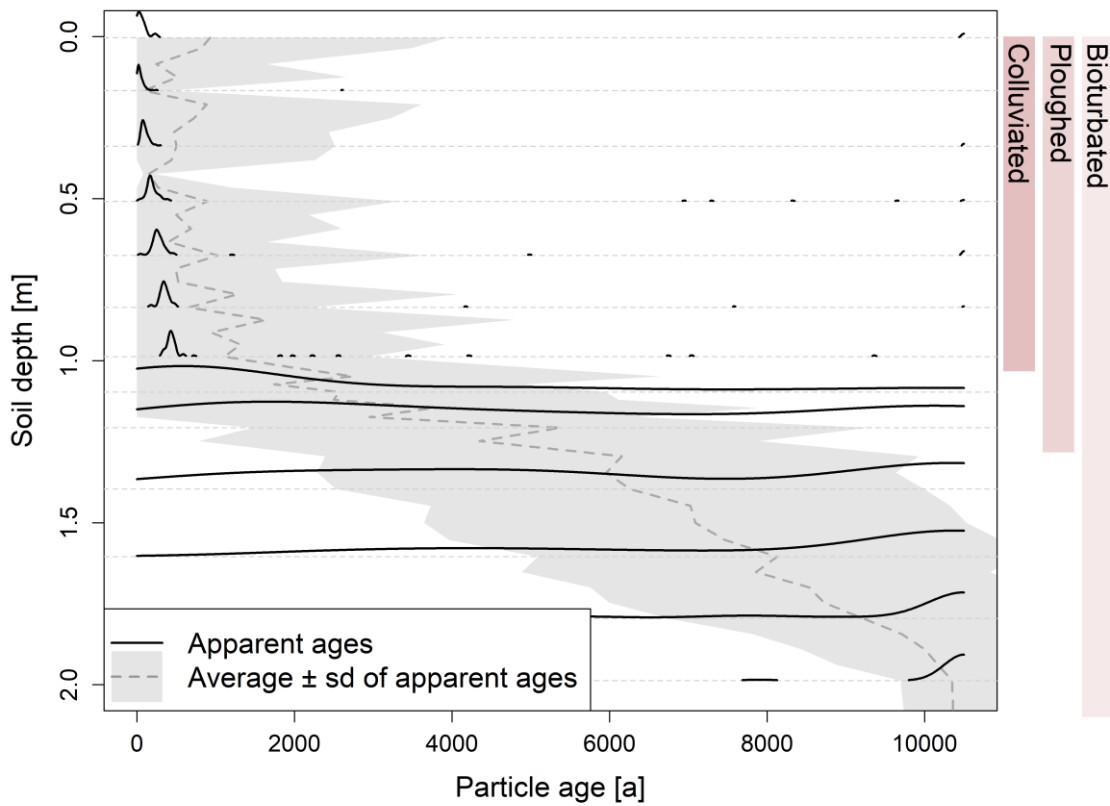

**Figure 4: Detailed age information for the depositional location. Age distributions of OSL particle ages are provided, as well as the**
**average and standard deviation of the OSL particle ages. The columns on the right indicate over which depths different processes**
**affected the OSL particle ages.**





Figure 4 shows a detailed age-depth profile at the depositional location. Under the former soil surface, around 1 meter, the age-depth profiles globally follow the age-depth profile created by bioturbation. There is no large rejuvenation in the top layers of this former soil, although these have also been tilled. Due to the high erosion and deposition rates by tillage, only a small
part of the particles in this plough layer had the opportunity to bleach, before they were buried under new colluvium. The incoming colluvium already contained a larger number of bleached particles, due to pre-bleaching at their erosion location and during transport. Therefore, the colluvial layer, starting at ~1 m, contains a high number of bleached particles. Nonetheless, there is still a number of particles with higher ages, even some which haven't been bleached.

Figure 5 show the development and changes in the cosmogenic nuclide inventories along the hillslope. In the natural phase,
the inventories of both meteoric and in-situ $^{10}$Be develop rather homogeneously and linearly along the hillslope, with only minor deviations due to the minor elevation changes by soil creep (Figure 5B&D). The linear increase over time shows that the inventories are still far from equilibrium, as can be expected by the simulated timescales. Both types of $^{10}$Be develop similarly, although with different magnitudes.

In the agricultural phase, the elevation changes drastically, due to the introduction of tillage erosion (Figure 5A). This also
affects the $^{10}$Be inventories (Figure 5C&E). The inventories show very different dynamics and rates of change compared to a natural landscape. The changes in inventories correspond closely to the elevation changes, indicating that the erosion and deposition are the major drivers of these changes. Differences in redistribution patterns of meteoric and in-situ $^{10}$Be can be attributed to variations in transport of different particle size fractions, caused by vertical heterogeneity developed in the natural phase. The largest changes in meteoric $^{10}$Be in the agricultural phase (peaks in Figure 5C), slowly move downslope with time.
This can indicate progressive erosion and transport of Bt horizons and their meteoric $^{10}$Be inventories along the hillslope.




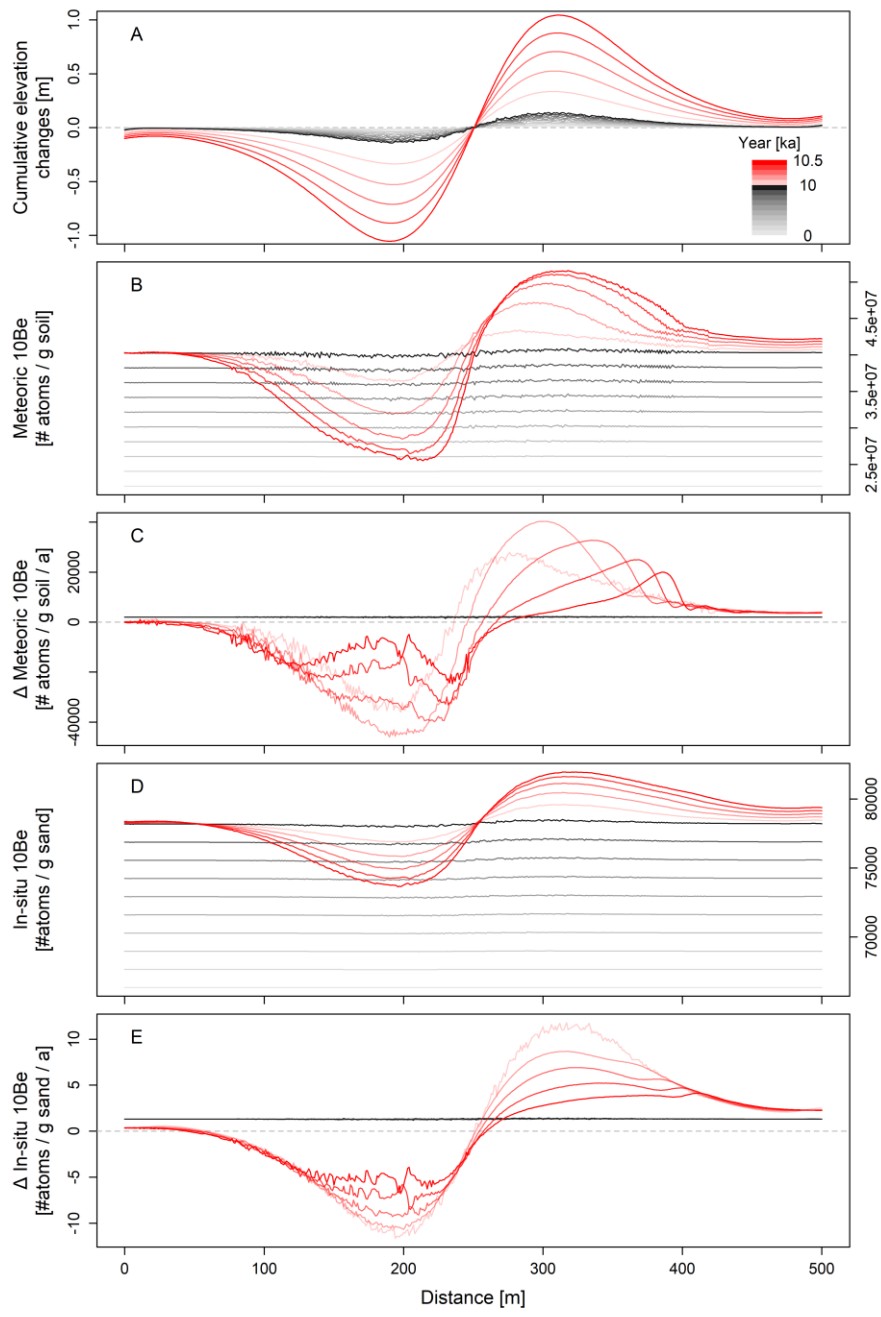

**Figure 5: Changes in elevation and cosmogenic nuclide inventories along the hillslope. The cosmogenic nuclide inventories show the total inventories for each soil profile. Grey colours indicate the natural phase. Red colours indicate the agricultural phase. Timesteps in between the results for the natural phase are 1000 years, and for the agricultural phase 100 years. The dashed lines in A, C and E indicate zero change. A: Changes in elevation. B: Changes in meteoric $^{10}$Be inventories. C: Rate of change in meteoric $^{10}$Be inventories. D: Changes in in-situ $^{10}$Be inventories. E: Rate of change in in-situ $^{10}$Be inventories. For C and E, differences in changes in the natural phase are very minor compared to the agricultural phase and not visible on this scale.**



## 5    Discussion

### 5.1    Simulated development of chronologies

The simultaneous simulation of soil and landscape evolution and the development of chronologies can provide new insights into the processes that form and affect the chronologies. In this Section, we discuss the simulated vertical, lateral and temporal distributions of geochronometers. We compare the simulations to observed profiles and add suggestions how simulations with ChronoLorica can support experimental studies on soil processes. Recall that all processes in this model run were uncalibrated. The parameters were chosen based on representative literature or chosen to create patterns in the outputs that could be expected

based on similar studies. The coupling of Lorica with a geochronological module enables the calibration of its processes using measured chronologies, but that is outside of the scope of this introductory paper.

### 5.1.1    Depth profiles

Depth profiles of cosmogenic nuclides and OSL particle ages help to understand the soil processes that are responsible for the development of these profiles (e.g., Graly et al., 2010; Johnson et al., 2014; Gray et al., 2020). The other way around,

simulation of these processes can help us understand which preconditions and rates are required to develop depth profiles that can be observed using field and experimental data. The simulated OSL particle age- and $^{10}$Be-depth profiles (Figure 2) resemble observed depth profiles. For example, the bulge shape of the meteoric $^{10}$Be resembles profiles observed in French loess soils (Jagercikova et al., 2015). Also for the bulge shape of meteoric $^{10}$Be that developed in the colluvium of the depositional profile, field analogues can be found (VAMOS profile in Calitri et al., 2019). Jagercikova et al. (2015) simulated the development of

their observed profiles using an advection-diffusion equation, on which Lorica's clay translocation process is based (see Section 2.2). This shows how experimental data help develop modelling tools for simulating the development of the observed profiles. When observations and simulations match well, the simulations can be expanded with additional processes or to other landscape positions, to understand how these extra factors influence the developments of chronological markers. For these locations, experimental approaches might not be applicable, because erosion processes can disturb the chronologies and

complicate their interpretation. Numerical simulations also provide the opportunity to test how different process rates, initial and boundary conditions affect the shape of the depth profiles (e.g., bulge, decline or uniform shapes, Graly et al., 2010), without the need to find suitable field analogues.

The simulations in the natural phase also show characteristic depth profiles for OSL particle ages and in-situ $^{10}$Be (Figure 2), which are affected by bioturbation (Wilkinson and Humphreys, 2005; Johnson et al., 2014; Román-Sánchez et al., 2019a).

The decreasing OSL particle ages towards the soil surface are a direct consequence of mixing by bioturbation. The rate at which bleached particles are mixed into the soil, depends on two factors: the mixing rate and the bleaching depth. The mixing rate is an obvious factor; with more intensive mixing, particles can reach quicker and deeper into the soil. The bleaching depth is a parameter that determines the supply of bleached particles. A larger bleaching depth supplies more bleached particles. Experimental data on bleaching depths in soils are scarce, and often numerical models are used to estimate these depths



(Furbish et al., 2018b). Factors that probably affect bleaching depths in soils are soil type and texture, vegetation cover and surface roughness affected by land use. To further improve numerical modelling of OSL ages in soils, additional experimental data on bleaching depths is required.

        Soil mixing in the upper part of soil profiles has been hypothesized and observed to affect radionuclide-depth profiles (Schaller et al., 2009; Hippe, 2017). This is also visible in our simulations (Figure 2D), where the upper part of the in-situ $^{10}$Be
is deviating from the expected exponential curve. The topsoil doesn't show a uniform concentration of in-situ $^{10}$Be, as suggested in Hippe (2017), because the soil is not completely mixed, as happens with tillage, and because mixing rates decrease with distance from the surface. This mixing pattern is typical for many natural soils (Gray et al., 2020), and can be used to quantify mixing rates (Wilkinson and Humphreys, 2005). Comparison of observations and model results using various ways and rates of simulating bioturbation will provide new insights in the applicability of cosmogenic nuclides for soil mixing
studies. However, as Hippe (2017) remarks, there is still a lack of field data for supporting such studies.

        Mixing by tillage is a much more intensive mixing process than bioturbation, creating uniform cosmogenic nuclide inventories and age distributions in the plough layer, as suggested by Hippe (2017). Tillage also mixes sediments that already have been deposited. This post-depositional mixing disturbs the depositional chronology and additional analysis is required to extract the required age information (Figure 4, Van der Meij et al., 2019). The average OSL particle ages do not match with
the modes of the age distributions (Figure 4), due to the presence of some older or unbleached grains. Just as with analysing partly bleached sediments from the field (Arnold et al., 2009; Cunningham and Wallinga, 2012; Van der Meij et al., 2019), also for the model results a minimum age model can be required to extract the required age information for comparison with field data.

### 5.1.2    Lateral redistribution patterns and rates

Next to soil development at a pedon scale, simulations with ChronoLorica also show how soils, landscapes and chronologies can evolve on a hillslope to landscape scale. This will help to understand where and how chronologies can form. This can assist in sampling site selection, testing hypotheses for landscape evolution (Crusius and Kenna, 2007), or calculation of erosion and deposition rates through model calibration (Temme et al., 2017).

        The effect of soil creep in the natural phase is limited with our parameter set, with elevation changes ranging from -0.12 to
+0.11 m in 10ka years, which corresponds to rates of -11 $*10^{-3}$ to +12$*10^{-3}$ mm a$^{-1}$. Creep rates reported for temperate and tropical environments range from 0.5 – 10 mm a$^{-1}$ (Saunders and Young, 1983), although the slopes of the measured rates ( 0 to >25°) are in general steeper than the ones in the simulations (average 4.5°, max 14°, Figure 1). Nonetheless, the measured rates are two to three orders of magnitude higher than the simulated rates. We used a conservatively estimated rate based on reported sediment fluxes by different soil biota (Gabet et al., 2003). The difference in rates might be explained by Eq. 1, where
the potential creep rate is multiplied with the slope gradient, which reduces the local rate substantially on our gentle sloping landscape. Tillage rates are much higher than the creep rates, ranging from -2 to +2 mm a$^{-1}$. These simulated tillage rates on gentle slopes thus equal natural soil creep rates on steep slopes. The simulated tillage rates fall in the range of reported average





agricultural erosion rates, although these reported rates show a very large spread (~0.1 - 10 mm a$^{-1}$ for 95% of the reported values, Montgomery, 2007). To further understand these geomorphological processes in real-world settings, calibration with field data is required.

The simulated chronologies show which geochronological methods are applicable for different landscape positions and over different timescales. For the cosmogenic nuclides, comparison between stable, eroded and deposition locations provide information on erosion and deposition rates (Figure 3; Phillips, 2000; Willenbring and von Blanckenburg, 2010; Calitri et al., 2019)). At the deposition location, the built-up colluvium contains a stratigraphic record of OSL particle ages (Figure 4), that can be used for determining deposition rates using conventional OSL measurements, although age corrections might be necessary to correct for post-depositional mixing processes (Van der Meij et al., 2019). At the stable or eroded position, there is no chronology that can be measured to determine erosion rate. However, as Figure 3 shows, the truncation of the OSL particle age profile is similar to the truncation of the $^{10}$Be profiles. This suggests that quantitative erosion rates can be derived by comparing eroded and stable bioturbation-age profiles, similar to truncation of radionuclide profiles (Arata et al., 2016a, b) or soil profiles (Van der Meij et al., 2017).

When considering the timescales of application, OSL ages provide numerical ages of different sediment layers, which enable the calculation of deposition rates in between different samples. The temporal resolution of OSL ages depends on the degree of bleaching, the amount of particles measured in an aliquot and the deposition rate (Dietze et al., 2022). When the deposition rate is high, a sample of a standard size will contain a large range of ages, compared to a low deposition rate. Cosmogenic nuclides provide averaged erosion and deposition rates over longer timescales. These timescales depend on the half-life of the nuclide and the dynamics of the landscape (Schaller and Ehlers, 2006; Mudd, 2017). Due to their often-long half-lives, cosmogenic nuclides can provide a pre-anthropogenic benchmark to compare to recent erosion rates (Kirchner et al., 2001). There are various ways to increase the temporal resolution of erosion rates determined with radionuclides. Fallout radionuclides, produced by nuclear bomb testing and nuclear accidents, can be used to determine recent erosion and deposition rates over the last ~70 years (Evrard et al., 2020). At this stage, fallout radionuclides are not implemented in ChronoLorica, but this can easily be done when timeseries of deposition are available. Cosmogenic nuclides with different half-lives can be combined to identify transience on relatively short timescales (e.g., Mudd, 2017; Hippe et al., 2021). For the combination of in-situ $^{14}$C and $^{10}$Be, accelerations in erosion rates can be detected starting ~500 years after the change (Mudd, 2017). A final approach is using numerical modelling to test various hypotheses of the development of radionuclide profiles (Vandermaelen et al., 2022).

ChronoLorica can be a valuable tool for evaluating and comparing different geochronometers. The simulations form a controlled experiment, with known rates of landscape change. The simulated changes in geochronometers can be used to evaluate their spatial and temporal resolution, compare different geochronometers and test analytical methods for determining erosion and deposition rates.



## 5.2 Limitations of ChronoLorica

ChronoLorica has several advantages over other numerical methods for simulating chronological development. These are its applicability for both steady state and transient landscapes, the possibility to simulate multiple geochronometers and the possibility to simulate various processes, including secondary processes such as post-depositional mixing. The previous sections have highlighted and illustrated these advantages. Here, we discuss model limitations.

### 5.2.1 Model uncertainties

Uncertainties in ChronoLorica mirror those in most soil-landscape evolution models. These relate to process formulations, initial and boundary conditions and data limitations (Minasny et al., 2015).

Lorica and ChronoLorica are developed mainly for Holocene and Anthropocene soil and landscape development, where changes in soils occur at similar rates as changes in the landscape. The architecture and process descriptions of the model were adjusted to these long timescales, with simplified process descriptions. The model should not be applied to shorter time scales (sub-annual to several years), because over these timescales, changes in soils, sediments and chronologies occur episodically. This behaviour is not captured in the simplified process descriptions, which simulate gradual changes over time. The model can be applied over longer timescales than the Holocene, but additional development might be required to include processes acting on these timescales and their effect on chronologies, for example various weathering processes. The architecture of ChronoLorica lends itself well for these adjustments, but extra care should be given to increased runtime and calculation demands of the model (see next Section).

All SLEMs face uncertainties coming from uncertain initial and boundary conditions. Over the simulated timescales, it is usually difficult, if not impossible, to make accurate reconstructions of the shape of the initial landscape, the properties of the parent material and climatic and anthropogenic changes over time (Minasny et al., 2015). A way to reduce or bypass these uncertainties is by performing simulations on hypothetical landscapes, as is done in this paper. This gives insights into how soils, landscapes and chronologies might react to different processes, rates and changes in boundary conditions. This can help to better understand the development of real-world landscapes. When these real-world landscapes are the topic of interest, the simulations are still dependent on local or regional reconstructions of boundary conditions. These reconstructions are often made using different chronological methods, such as pollen analysis and [14]C dates for climate and vegetation reconstruction (Mauri et al., 2015), or OSL and other dating methods for regional land use history and landscape change (e.g., Kappler et al., 2018, 2019; Pierik et al., 2018). These reconstructions serve as input for SLEMs, but, interestingly, SLEMs such as ChronoLorica can also be used to better understand the chronologies that have been used for developing these reconstructions.

### 5.2.2 Runtime and memory constraints

The runtime of the model, i.e. the time it takes to finish a simulation, depends on the vertical and temporal discretization of the model scenario: raster dimensions and cell size, number of soil layers and the number of time steps in the model. The





runtime increases supralinearly with the dimensions of the soil landscape. For instance, for bioturbation, the number of calculations increases exponentially with the number of soil layers, because there is exchange between each soil layer and all other layers. The simulations for this paper, with a raster of 1 by 501 cells, 25 soil layers and 10500 simulation years took nearly twenty hours on an average (year 2022) laptop.

The spatial and temporal dimensions of a simulation thus need to be chosen with care to limit the runtime of a simulation. For two-dimensional landscapes, the cell size of the input raster can be increased to reduce the number of raster cells. The thickness of the soil layers can also be adjusted. ChronoLorica provides the option of varying soil layer thickness, where layers closer to the surface are thinner than subsurface layers. This provides more detail in the zone where most variation is expected. When choosing the number and thickness of soil layers, the vertical range of different processes should also be considered.

For example, when the plough depth is set to 25 cm, the layer thickness should ideally be much smaller to prevent the plough layer from mixing layers that is partially located in the plough depth. The layer thickness of 5 cm in our simulations is relatively large, but was chosen to limit the calculation time.

From the geochronological module, runtime is substantially increased by the particle tracing. With each movement of sediments, either laterally by geomorphological processes or vertically by soil processes, there is a probability that the particles

in the source layers move as well. For each individual particle, the model probabilistically assesses whether it moves together with the sediment. This requires a lot of extra calculation steps. In comparison, radionuclide inventories require only one extra calculation step, as a fraction of the inventories is moved between the layers.

The choice of the numbers of particles per layer should depend on three factors: the spatial and temporal discretization of the model, the simulated processes and the sand content of each layer. For a one-dimensional simulation of soil profile

development, for example by bioturbation, much more particles can be simulated in the same calculation time as a full three-dimensional landscape. Bioturbation requires a lot of calculation time, especially when the OSL particle age module is activated. When simulating three-dimensional landscapes, it is wise to consider whether bioturbation has a large effect on the chronologies compared to the other processes. If not, the exclusion of bioturbation simulation can be considered. The final consideration is the sand content of each layer, as the particles are associated with sand fraction. It is important to choose the

number of particles in a way that the bleaching layer has at least one or two particles present. Otherwise, there is a chance that particles are not bleached in the model run, or that the distribution of OSL particle ages does not provide usable information. A way to estimate the number of particles in the bleaching layer is multiplying the bleaching depth with the cell size, an average bulk density (e.g., 1500 kg m$^{-3}$) and sand fraction. This gives the sand content in the bleached layer, which can be used to estimate the initial number of particles per kilogram of sand. In this paper, this results in 0.005 * 1 * 1 * 1500 * 0.25 = 1.9

kg sand. With four particles per kilogram of sand, this results in ~8 particles in the bleached layer.

Increasing spatial and temporal dimensions also influences the memory requirements of the model and the size of the output data. The output of the simulations for this paper, on a one-dimensional hillslope, is 6 GB in size, with outputs every 100 years of the simulations. 86% of these data are output files containing the OSL particle age information. Constraining the spatial and



temporal dimensions of the simulations will also help to constrain the memory requirements for the simulations and will help
speed up the analysis of the model output.

We propose the following workflow when applying the model. First, run the simulations without geochronological module, to get an understanding of spatial and temporal variation in soil and landscape properties. Based on that, the spatial discretization can be chosen. Second, determine the number of particles per kilogram of sand, using the guidelines described above. Last, run the model with geochronological module to get the required age information.

### 5.3   ChronoLorica for other pedogenical, geomorphological or geological applications

The current version of ChronoLorica was developed with agricultural landscapes in mind, because these transient landscapes show the highest rates of landscape change, which are difficult to measure with conventional methods (Calitri et al., 2019). ChronoLorica can easily be adapted for other landscapes, settings and processes. In this Section we mention several possible applications and the required adaptations to the model.

#### 5.3.1   Calibration of bioturbation processes and rates

Bioturbation, the process where soil biota mixes the soil, affects the redistribution of geochronometers as well (Wilkinson and Humphreys, 2005). By confronting ChronoLorica with experimental data, the bioturbation rate and depth decay rate can be calibrated. ChronoLorica can also be used to test various formulations of the bioturbation process. Currently, we simulate bioturbation as a process where the entire profile is prone to mixing, with decreasing rates further from the surface. This
process resembles soil mixing by earthworms. Other process descriptions, such as upward transport of particles, for example done by termites (Kristensen et al., 2015), or instantaneous mixing of a soil body by tree uprooting (Šamonil et al., 2015) can easily be implemented in the model and its effect on chronologies simulated. This flexibility facilitates hypothesis testing for determining which bioturbation process might have been responsible for an observed depth profile.

#### 5.3.2   Landscape evolution modelling on soil-mantled hillslopes

A traditional application of landscape evolution models (LEMs) is to understand how soil-mantled hillslopes evolve. Over geological timescales, LEMs assume that there is a balance between soil production and soil erosion, either by water or by diffusive processes (Tucker and Hancock, 2010). Cosmogenic nuclides are a common method to study this landscape development as well, by calculating spatially varying or catchment-averaged erosion rates (Bierman and Steig, 1996; Granger et al., 1996). Most LEMs consider soils as a mobile part of the hillslope and do not subdivide soils in multiple layers or simulate
vertical transport among layers. This complicates the comparison with measured cosmogenic nuclide-depth profiles with simulations. ChronoLorica can support these studies, by simulating the cosmogenic nuclide-depth profiles or OSL age-profiles (Figure 3) that can be compared to field measurements.

For these studies, soil processes are of minor importance and focus can be placed on the geomorphological processes. Required adjustments to the model code include the development of a geochronological module for the bedrock weathering




process. Special care should be given to how cosmogenic nuclides are distributed initially in soils and bedrock, and how weathering changes the bulk density and cosmogenic nuclide inventory upon the conversion of bedrock into soil.

### 5.3.3 Particle-size selectivity in mixing and transport processes

ChronoLorica considers different grain sizes: coarse, sand, silt, clay and fine clay. Grain size controls the uptake and deposition by water, or the rate of physical and chemical weathering. Other soil processes in Lorica, such as bioturbation, are
currently grain-size independent, although there is evidence that also bioturbation can be grain-size dependent (Dashtgard et al., 2008). Empirical studies measuring geochronometers associated with different grain sizes can help the formulation of a grain-size dependent soil mixing module for ChronoLorica, which in turn can help to determine mixing rates for different particle sizes and improve simulation of age distributions. With small adjustments, ChronoLorica can consider a larger range of particle sizes, providing more detail in the simulated processes.

### 5.3.4 Soil weathering effects on cosmogenic nuclide distributions

Weathering is the breakdown of coarser particles into smaller particles. This can be due to physical processes, such as free-thaw cycles, or due to chemical dissolution processes.

Chemical weathering can lead to quartz enrichment by removing other minerals. This can overestimate in-situ cosmogenic nuclide production (Riebe et al., 2001), change cosmogenic nuclide inventories from weathering bedrock into finer soil
fractions (Ott et al., 2022), or even promote the bleeding of in-situ [10]Be into meteoric [10]Be pools. With addition of a chronological module to the different weathering processes in ChronoLorica, the effects of weathering on cosmogenic nuclide distributions can be quantified.

### 6 Conclusions

ChronoLorica is a coupling between soil-landscape evolution model Lorica and a geochronological module, which traces
various geochronometers throughout the simulations of soil and landscape development. The model simulates realistic spatial and temporal patterns of the geochronometers, under both natural and agricultural land use. It simulates these patterns over large spatial and temporal extents, and therefore provides rich possibilities for data-based calibration. By combining different geochronometers, the model can be applied in both steady-state and transient landscapes, where quickly changing boundary conditions, such as land management intensification and climate change, increase rates of landscape change and create complex
geo-archives. The flexible framework of ChronoLorica can easily be expanded with other geochronometers and processes, which facilitates its deployment in different pedological, geomorphological and geological applications.



**Code availability**

The model is accessible via https://github.com/arnaudtemme/lorica_all_versions/releases/tag/1.0.

**Author contributions**

MvdM and AT developed the model code. SB and TR contributed the geochronological information. MvdM performed the simulations and analysed the results. MvdM prepared the manuscript with contributions from all co-authors.

**Competing interests**

The authors declare that they have no competing interests.





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
