# Peer review of "ChronoLorica – Introduction of a soil-landscape evolution model combined with geochronometers"

_EGUsphere, 2022_

## Referee Comment (RC1)

The manuscript entitled "ChronoLorica – Introduction of a soil-landscape evolution model combined with geochronometers" by van der Meij et al. presents a novel model to simultaneously simulate soil and landscape evolution, respectively. This contribution starts filling an important knowledge and tool gap. To the best of my knowledge, such models commonly simulate either landscape or soil evolution but only rarely both. I therefore highly appreciate this contribution. To this end, the authors combine lateral matter fluxes, i.e. diffusion, advection, to simulate hillslope formation with vertical processes that shape the soil evolution, i.e. bioturbation, clay translocation. I enjoyed reading the manuscript a lot and I think this manuscript nicely suits the focus of GChron. Moreover, I think the presented modeling here will find use not only in geochronology community but will raise attention among both soil scientists and geomorphologists. In fact, this study could have a high impact. One of the most interesting and potentially impactful implications of this study is arises late in the discussion (L 548ff: 'These [reconstructions] are often made using different chronological methods, such as pollen analysis and $^{14}$C dates for climate and vegetation reconstruction (Mauri et al., 2015), or OSL and other dating methods for regional land use history and landscape change (e.g., Kappler et al.,2018, 2019; Pierik et al., 2018). These reconstructions serve as input for SLEMs, but, interestingly, SLEMs such as ChronoLorica can also be used to better understand the chronologies that have been used for developing these reconstructions.' The chosen approach is plausible and the implementation into a freely available software provided on Github is consistent. The English is well written and the figures are clear and easy to follow.
However, I have several major observations that I wish to address to the authors.

(1) The authors present a first and single simulation of soil and landscape co-evolution. To this end, the authors chose a (thankful) example of a synthetic, sigmoidal-shaped hillslope that is based on the shape diffusion dominated. While I clearly see the scope of this study and I also acknowledge the aim of a 'proof-of-concept', I was wondering why the authors chose such a hillslope shape and did not test for other hillslope shapes that are not as much controlled by diffusive processes. I was wondering if the model can perform on distinctly shaped hillslopes comparably well?

(2) I missed a detailed description of the boundary conditions applied to the modeling raster. I assume that the hillslope extends from x = 0 at the ridge to x = L at the valley bottom. I anticipate that the boundary conditions are set to $\frac{\partial z}{\partial x}|_{(o,t)} = 0$ and $z(L,t) = 0$. Could the authors please provide more information on the boundary conditions. Also, a description of the initial conditions would be appreciated and may improve the readability of the manuscript allowing reproduction.

(3) In this context, I also miss a description and reasoning for the parameters chosen (Table 1). I understand that the aim of this study is a proof-of-concept. However, I cannot see any explanation of how the estimated parameters are chosen. I see this as an important gap given that in L 434 the parameters are presumably chosen 'to create outputs that could be expected'. This may introduce a bias.

(4) The authors state that the current model includes several important geomorphological processes including tree throw. I missed the application here, as this process is prominently mentioned in line 110-111. I did not find this specific process in the Github repository neither. Any chance to include this process into the current study, too?

(5) Given the clearly stated focus of the study that restricts on proof-of-concept and does not claim to reconstruct existing topography and soil landscapes, I was wondering if the authors should not, however, better context their modeling approach into the 'real world'. Any idea of how plausible (in quantitative terms!) the model may simulate existing landscapes?

(6) Generally, I see problems in the organization of the results and discussion section. In many occasions, e.g. LL, the authors mix the results with an interpretation under the umbrella of a results section, e.g., L357 'indicating' or L370 'This is a consequence of...' . I would recommend to more rigorously split results and discussion. Given the current version, I have

problems in objectively assessing the results. In addition, I miss more quantitative statements of the results. In many cases the authors remain unclear by stating 'more than' etc, e.g. L 385 'the inventories are higher compared to…' or L 415 'show very different dynamics…'

(7) I see a conflict in the modelled rates of vs measured rates of erosion. How do the authors explain the difference of 2-3 orders of magnitude between observed and modeled values excluding tillage as the process responsible? If I understood the manuscript correctly, such high discrepancy also occurs under presumably 'undisturbed' conditions during the 'natural phase'. Thus, I am not sure if comparing these data with tillage is plausible. The other explanation of the potential creep rate as multiplied by the slope gradient needs more explanation at least.

(8) Also, the authors claim a probabilistic approach for choosing the particles. Yet, there are no clear descriptions of how the probability is computed. I see the reasoning of the fractions of sand etc. Yet, this is not unambiguously clear here how the probability (that is not the fraction) is estimated here.

(9) How do the authors define here transient landscapes? Do the authors refer here to the fact that the modeled curves of hillslope elevation still evolve over the period of simulation without convergence? Or do the authors refer here to landscapes changing in terms of erosion processes, i.e. tillage is activated after some 'natural' and undisturbed periods? Or are the boundary conditions changing over time. In that case, I suggest to be more explicit and I refer to my comment above.

(10) I personally liked the clear, fair and honest discussion on the 'weaknesses' of the model. Yet, in the way it is written now, I have had the impression that many 'easy to apply' ('easy' is commonly mentioned here) additions to the model can be achieved. If so, and if these extensions are that easy, why aren't they already implemented in the current model version?

(11) Finally, I think a sensitivity analysis of the parameters (and thus the underlying processes) would improve the manuscript a lot. Up to now, it is hard to assess the efficacy of the distinct processes implemented on the simulated patterns. Such an addition, which is a lot of work, I know, may help to disentangle quantitatively the impact of controls on the simulated results.

Minor
(1) I suggest to avoid words like 'complex'. Every landscape is complex. What does complex refer here to?

(2) LL 127f I did not fully got the 'division of the slope gradient' and 'factor p'. Maybe the authors could better describe here the procedure. Similarly, L 142 on the 'convergence' factor.

(3) L 153: 'is lost from the soil column'. How? Here, again the boundary conditions are important. How can soil be lost assuming the conservation of mass? I assume that this principle applies here, too as loss and gain in elevation equals 1.04 m in both cases. If this is not the case, please state clearly.

(4) L 262-265: This sentence is hard to understand. Please consider rephrasing.

(5) Table 1: What is LSD in the table exactly? I missed an explanation here.

(6) L 503. 'This suggests..' I did not fully understood this sentence. Please consider rephrasing.

(7) The paragraph LL506-520 reads a bit out of context. I suggest to better connect this section to the discussion of the results obtained by modeling.

(8) L 559 Please provide more specifics on the computing infrastructure. A laptop of year 2022 can be anything.

---

## Author Response (AR1)

Dear editor,

Thank you for handling our submission to Geochronology. We have finished the revised manuscript, where we addressed all of the reviewer's comments. I added our responses to each referee comment below. Here, we list the most important changes to the manuscript. The line numbers refer to the revised manuscript.

- We added a (better) motivation and explanation for certain model settings, parameters and boundary conditions. These include:
  - The exponential depth curve for various processes (lines 129 – 135);
  - Mass conservation in the model (lines 108 – 110);
  - Boundary conditions of the input elevation transect (lines 327 – 332);
  - Parameters that we estimated based on comparable studies or desired model behaviour (lines 344 – 354).
- We moved interpretative statements from the Results section and incorporated them in the Discussion where suitable (lines 475 – 484)
- We added a potential application of the model for studying the effect of spatiotemporal variation of dose rates on OSL ages (Section 5.3.2)

With the changes in the manuscript, we think that the manuscript has improved and is easier to understand for the reader. We hope it is now suitable for publication.

On behalf of all authors,

With best regards,

Marijn van der Meij

**Reply to anonymous referee 1**

*The manuscript entitled "ChronoLorica – Introduction of a soil-landscape evolution model combined with geochronometers" by van der Meij et al. presents a novel model to simultaneously simulate soil and landscape evolution, respectively. This contribution starts filling an important knowledge and tool gap. To the best of my knowledge, such models commonly simulate either landscape or soil evolution but only rarely both. I therefore highly appreciate this contribution. To this end, the authors combine lateral matter fluxes, i.e. diffusion, advection, to simulate hillslope formation with vertical processes that shape the soil evolution, i.e. bioturbation, clay translocation. I enjoyed reading the manuscript a lot and I think this manuscript nicely suits the focus of GChron. Moreover, I think the presented modeling here will find use not only in geochronology community but will raise attention among both soil scientists and geomorphologists. In fact, this study could have a high impact. One of the most interesting and potentially impactful implications of this study is arises late in the discussion (L 548ff: 'These [reconstructions] are often made using different chronological methods, such as pollen analysis and 14C dates for climate and vegetation reconstruction (Mauri et al., 2015), or OSL and other dating methods for regional land use history and landscape change (e.g., Kappler et al.,2018, 2019; Pierik et al., 2018). These reconstructions serve as input for SLEMs, but, interestingly, SLEMs such as ChronoLorica can also be used to better understand the chronologies that have been used for developing these reconstructions.' The chosen approach is plausible and the implementation into a freely available software provided on Github is consistent. The English is well written and the figures are clear and easy to follow. However, I have several major observations that I wish to address to the authors.*

**Response:** Dear referee,

Thank you for the kind words and thorough review of our manuscript. Soil-landscape evolution models (SLEMs) indeed fill the gap between one-dimensional soil evolution models and two-dimensional landscape evolution models. However, the model we present is not one of the first models to couple soil evolution and landscape evolution. In Minasny et al., (2015) and Van der Meij et al., (2018), several SLEMs are reviewed, including the SLEM Lorica (see also lines 60-62 in the manuscript). The model we present is an extension to Lorica, where we added the geochronological modules (line 96).

Below, we address your remarks one by one.

With best regards, on behalf of all authors,
Marijn van der Meij

> (1) *The authors present a first and single simulation of soil and landscape co-evolution. To this end, the authors chose a (thankful) example of a synthetic, sigmoidal-shaped hillslope that is based on the shape diffusion dominated. While I clearly see the scope of this study and I also acknowledge the aim of a 'proof-of-concept', I was wondering why the authors chose such a hillslope shape and did not test for other hillslope shapes that are not as much controlled by diffusive processes. I was wondering if the model can perform on distinctly shaped hillslopes comparably well?*

**Response:** The model also works well on distinctly-shaped hillslopes, as is illustrated in earlier publications of the Lorica model (e.g., Temme and Vanwalleghem, 2016; Van der Meij et al., 2020). In these landscapes, diffusive processes act in two dimensions instead of on a one-dimensional hillslope, and advective geomorphological processes, such as water erosion can be simulated as well. We chose to simulate the one-dimensional sigmoidal-shaped landscape for two reasons. The first is that the simulation results for such a simplified landscape are easy to visualize and explain, which serves the purpose of this introductory paper of the model. The second is that advective processes in the model, such as water erosion and deposition, rarely lead to the development of depositional layers in the simulations, which is not useful to illustrate the development of chronologies in depositional environments. We will mention these reasons in Section 3 of the manuscript. The application of the model in distinctly-shaped landscapes, that are formed under different processes, is something that we are planning for future publications.

> (2) *I missed a detailed description of the boundary conditions applied to the modeling raster. I assume that the hillslope extends from x = 0 at the ridge to x = L at the valley bottom. I anticipate that the boundary conditions are set to $\frac{\partial z}{\partial x}|_{(0,t)} = 0$ and $z(L,t) = 0$. Could the authors please provide more*

*information on the boundary conditions. Also, a description of the initial conditions would be appreciated and may improve the readability of the manuscript allowing reproduction.*

**Response:** We will add the following initial and boundary conditions on the modelling raster to the manuscript: "The simulated hillslope was created to present stable, eroding and depositional positions under conditions of diffusion and has the shape of a Gaussian curve. The hillslope extends from $x = 0$ m at the ridge to $x = 500$ m at the valley bottom. $z(0,0) = 40$ m and $z(500,0) = 0$ m. Through the simulations, $z(x,t)$ changes under the influence of the simulated pedological and geomorphological processes. There are no restraints on $\partial z(x,t)$. "

(3) *In this context, I also miss a description and reasoning for the parameters chosen (Table 1). I understand that the aim of this study is a proof-of-concept. However, I cannot see any explanation of how the estimated parameters are chosen. I see this as an important gap given that in L 434 the parameters are presumably chosen 'to create outputs that could be expected'. This may introduce a bias.*

**Response:** The selected parameters are loosely based on values that we got from literature and from previous modelling studies. The parameters are in the same order of magnitude as reported values in literature, but we did not want to add these references to Table 1, because we didn't use the exact same parameters. Other parameters we estimated to get illustrative outcomes for the model. An example is the tillage constant. With a too low value, the build-up of colluvium would be limited and we would not be able to illustrate how the model simulates the development of geochronometers in the colluvium.
However, we agree that the justification of the selected parameters is very limited. We will add an elaborated justification of all parameters to the text in Chapter 3. We will add the references on which we based our parameters and we will better explain what we mean with selecting parameters to get illustrative outcomes.

(4) *The authors state that the current model includes several important geomorphological processes including tree throw. I missed the application here, as this process is prominently mentioned in line 110-111. I did not find this specific process in the Github repository neither. Any chance to include this process into the current study, too?*

**Response:** The process is called *calculate_tree_fall()* in the github code (code lines 18016-18410). This process is included in the simulations in Van der Meij et al., (2020), where it was shown to be a major process causing soil heterogeneity. We did not include it in this study, because tree throw is not commonly studied using geochronological tools, and we deemed it not a representative process to illustrate our model. The geochronological module for this process is also not yet written.

To avoid this confusion, we will mention specifically which processes were included in the simulations in this study. We will list the other processes in the model separately, in case someone else wants to apply the model in a study area where other processes are occurring.

(5) *Given the clearly stated focus of the study that restricts on proof-of-concept and does not claim to reconstruct existing topography and soil landscapes, I was wondering if the authors should not, however, better context their modeling approach into the 'real world'. Any idea of how plausible (in quantitative terms!) the model may simulate existing landscapes?*

**Response:** The performance of SLEMs for simulating real-world landscapes depend on several aspects: the spatial and temporal extents of the simulations, the complexity of the actual soil and landscape evolution, and the data availability for calibrating the model and reconstructing initial and boundary conditions. Based on these aspects, the quality of the simulations will differ a lot between different landscapes and is difficult to estimate in advance. We also explained this in Section 5.2.1.

Actually, one of the motivations for developing the geochronological module for Lorica, was to improve the application of SLEMs for simulating real-world landscapes. The module provides an extra possibility for calibration and validation of the model. We will mention this again in Section 5.2.1, and we will also stress the importance of providing quantitative evaluation of the model when it is applied in real-world settings.

(6) *Generally, I see problems in the organization of the results and discussion section. In many occasions, e.g. LL, the authors mix the results with an interpretation under the umbrella of a results section, e.g., L357 'indicating' or L370 'This is a consequence of...'. I would recommend to more rigorously split results and discussion. Given the current version, I have problems in objectively assessing the results.*

*In addition, I miss more quantitative statements of the results. In many cases the authors remain unclear by stating 'more than' etc, e.g. L 385 'the inventories are higher compared to…' or L 415 'show very different dynamics…'*

**Response:** Thank you for this remark. We will adjust the Results section where necessary, to remove any interpretation. Concerning the remark about quantitative statements, in this paper we want to illustrate how the model works and how different processes affect different geochronometers. We think that relative statements, such as 'more than' or 'higher than' better serve this purpose than quantitative statements, because we want to show what happens in the model. Quantitative statements will be relevant when the simulated data is confronted or validated with field data, which is not the scope of this paper.

(7) *I see a conflict in the modelled rates of vs measured rates of erosion. How do the authors explain the difference of 2-3 orders of magnitude between observed and modeled values excluding tillage as the process responsible? If I understood the manuscript correctly, such high discrepancy also occurs under presumably 'undisturbed' conditions during the 'natural phase'. Thus, I am not sure if comparing these data with tillage is plausible. The other explanation of the potential creep rate as multiplied by the slope gradient needs more explanation at least.*

**Response:** The simulated creep rates in the natural phase are indeed much lower than measured creep rates in the field. In the manuscript, we provide several explanations, such as shallower slopes and conservatively estimated parameters in the simulations. The point that we want to make is that the model needs to be confronted with field data to improve creep simulations and derive better model parameters, among others with our geochronological module. We will stress this point more in the manuscript. We will remove the comparison of creep rates with tillage rates.

(8) *Also, the authors claim a probabilistic approach for choosing the particles. Yet, there are no clear descriptions of how the probability is computed. I see the reasoning of the fractions of sand etc. Yet, this is not unambiguously clear here how the probability (that is not the fraction) is estimated here.*

**Response:** The probability that a particle is transported, equals the fraction of sand that leaves a certain layer. So, if 1% of all sand is removed from a layer, also 1% of the particles should be removed, because they are associated with the sand fraction. Because the number of particles is variable, we need to calculate a probability for particle transport. This probability equals the fraction of sand that is transported, in this case $P = 0.01$. This probability is then used to randomly assess for each particle, if it is transported or not. We will better explain how we derive the transport probabilities in the manuscript.

(9) *How do the authors define here transient landscapes? Do the authors refer here to the fact that the modeled curves of hillslope elevation still evolve over the period of simulation without convergence? Or do the authors refer here to landscapes changing in terms of erosion processes, i.e. tillage is activated after some 'natural' and undisturbed periods? Or are the boundary conditions changing over time. In that case, I suggest to be more explicit and I refer to my comment above.*

**Response:** With transient landscapes, we actually mean all the aspects that the reviewer suggests. By changes in boundary conditions, for example land use intensification or introduction of tillage erosion, erosion causes landscapes, soils and geochronometers to change with rates that are much higher than natural changes, that are often convergent or steady-state. We will better define what we mean with transient landscapes in the Introduction.

(10) *I personally liked the clear, fair and honest discussion on the 'weaknesses' of the model. Yet, in the way it is written now, I have had the impression that many 'easy to apply' ('easy' is commonly mentioned here) additions to the model can be achieved. If so, and if these extensions are that easy, why aren't they already implemented in the current model version?*

**Response:** With Section 5.3 we want to indicate for what kind of scientific questions ChronoLorica could be used, and maybe inspire other researchers to use SLEMs such as ChronoLorica in their research. Adapting the model to answer these questions requires additional data and knowledge, that is not readily available to us. Also, each of these topics will need justification of the model adjustments and calibration and validation of the model results, which in itself can fill up an entire paper. Therefore, we think it is outside of the scope of this introductory paper to include the proposed adaptations. In hindsight, 'easy' is not the right word to refer to the model adaptations. We will leave out this word when we discuss the possible model adaptations in Section 5.3.

*(11) Finally, I think a sensitivity analysis of the parameters (and thus the underlying processes) would improve the manuscript a lot. Up to now, it is hard to assess the efficacy of the distinct processes implemented on the simulated patterns. Such an addition, which is a lot of work, I know, may help to disentangle quantitatively the impact of controls on the simulated results.*

**Response:** We agree that a sensitivity analysis can shed more light on the effect of the individual parameters on the simulated results. This was also shown in a sensitivity analysis for the original Lorica model (Temme and Vanwalleghem, 2016).

A sensitivity analysis of the geochronological module would be most useful when focusing on a single process instead of a collection of processes, because different processes require different settings on the geochronological module (see Section 5.2.2). We believe that quantitative analysis of parameter selection and sensitivity can best be preserved for future studies where we will focus on individual processes and where we can confront the model with experimental data, and is therefore outside the scope of this paper.

*Minor*

*(1) I suggest to avoid words like 'complex'. Every landscape is complex. What does complex refer here to?*
**Response:** With complexity, we mean that landscapes and geo-archives are formed by multiple processes, sometimes under non-linear behaviour. This complicates the disentanglement of the effects of individual processes. This is in contrast to 'simple' landscapes, where there is a dominant shaping process. The term complexity is commonly used in geomorphology (e.g., Temme et al., 2015) We will add this reference to the manuscript to indicate what we mean with complexity in this context.

*(2) LL 127f I did not fully got the 'division of the slope gradient' and 'factor p'. Maybe the authors could better describe here the procedure. Similarly, L 142 on the 'convergence' factor.*
**Response:** The part of the equation with the convergence factor $p$ determines diffusive transport through the landscape. We will add the following clarification to the text: "This last part of the equation controls the diffusive transport through the landscape, using the multiple flow algorithm (Freeman, 1991). The parameter $p$ determines the division of the transport over all lower lying neighbouring cells. With higher values of $p$, transport becomes more convergent towards the lowest neighbouring cell."

*(3) L 153: 'is lost from the soil column'. How? Here, again the boundary conditions are important. How can soil be lost assuming the conservation of mass? I assume that this principle applies here, too as loss and gain in elevation equals 1.04 m in both cases. If this is not the case, please state clearly.*
**Response:** Clay particles that eluviate from the lowest layer, are assumed to leach from the soil columns. This resembles leaching to deep layers, of colloidal transport. In modelling terms, these clay particles are removed from the modelling domain. This is necessary, because otherwise the clay particles will accumulate in the lowest soil layer, creating an unrealistic clay fraction. The conservation of mass still applies, when we consider this loss term. We will mention the boundary condition of conservation of mass in Section 2.1: model architecture.

*(4) L 262-265: This sentence is hard to understand. Please consider rephrasing.*
**Response:** We will rephrase this sentence to: 'The total local input, $A_{me,local}$, is divided over all soil layers at that location, based on the depth of the respective layer and the depth decay function (Eq. 11)'

*(5) Table 1: What is LSD in the table exactly? I missed an explanation here.*
**Response:** LSDn is a production rate scaling scheme used to normalize measured cosmogenic nuclide concentrations to globally distributed calibration sites, adjusting for variability in production rate with altitude and magnetic field influences. LSD is shorthand for Lifton, Sato and Dunai, the authors of the paper describing the scheme (Lifton et al., 2014). By accident we omitted the reference to this paper, so we will add it to Table 1.

*(6) L 503. 'This suggests..' I did not fully understood this sentence. Please consider rephrasing.*
**Response:** We will rephrase this sentence to: "This suggests that quantitative erosion rates can be determined by the level of truncation (i.e. 'decapitation' of depth profiles) of bioturbation age-depth profiles, similar to truncation of radionuclide profiles (Arata et al., 2016a, b) or soil horizon profiles (Van der Meij et al., 2017)."

*(7) The paragraph LL506-520 reads a bit out of context. I suggest to better connect this section to the discussion of the results obtained by modeling.*
**Response:** This Section is indeed out of context. We will leave it out of the next version of the manuscript, because it does not add to the Discussion of the model results.

*(8) L 559 Please provide more specifics on the computing infrastructure. A laptop of year 2022 can be anything.*

**Response:** We will add the specifics of the laptop that was used for the simulations to the manuscript (Intel Core i7 processor with 6 cores and clock speed of 2.7 GHz, 16 GB RAM).

**References**

Freeman, T. G.: Calculating catchment area with divergent flow based on a regular grid, Computers & geosciences, 17, 413–422, https://doi.org/10.1016/0098-3004(91)90048-I, 1991.

Lifton, N., Sato, T., and Dunai, T. J.: Scaling in situ cosmogenic nuclide production rates using analytical approximations to atmospheric cosmic-ray fluxes, Earth and Planetary Science Letters, 386, 149–160, https://doi.org/10.1016/j.epsl.2013.10.052, 2014.

Minasny, B., Finke, P., Stockmann, U., Vanwalleghem, T., and McBratney, A. B.: Resolving the integral connection between pedogenesis and landscape evolution, Earth-Science Reviews, 150, 102–120, https://doi.org/10.1016/j.earscirev.2015.07.004, 2015.

Temme, A. J. A. M. and Vanwalleghem, T.: LORICA – A new model for linking landscape and soil profile evolution: development and sensitivity analysis, Computers & Geosciences, 90, 131–143, https://doi.org/10.1016/j.cageo.2015.08.004, 2016.

Temme, A. J. A. M., Keiler, M., Karssenberg, D., and Lang, A.: Complexity and non-linearity in earth surface processes – concepts, methods and applications, Earth Surface Processes and Landforms, 40, 1270–1274, https://doi.org/10.1002/esp.3712, 2015.

Van der Meij, W. M., Temme, A. J. A. M., Lin, H. S., Gerke, H. H., and Sommer, M.: On the role of hydrologic processes in soil and landscape evolution modeling: concepts, complications and partial solutions, Earth-Science Reviews, 185, 1088–1106, https://doi.org/10.1016/j.earscirev.2018.09.001, 2018.

Van der Meij, W. M., Temme, A. J., Wallinga, J., and Sommer, M.: Modeling soil and landscape evolution–the effect of rainfall and land-use change on soil and landscape patterns, Soil, 6, 337–358, https://doi.org/10.5194/soil-6-337-2020, 2020.

**Response to the review of Harrison Gray**

*H. Gray, PhD*

*Research Geologist*

*U.S. Geological Survey*

*Van Der Meij et al. present an introduction to an adaption of an established soil landscape evolution model by including new processes controlling Optically Stimulated Luminescence (OSL) and cosmogenic geochronometers. The authors introduce this new model as a start into delving into broader scale questions of landscape dynamics.*

*Overall, I am very supportive of this paper. I think that coupling luminescence (and cosmogenics) into a large-scale landscape evolution model is a great idea. In particular, as the authors note, this approach has the potential to uncover new predictions and hypotheses that would be hard to develop outside of a modeling framework. Also it is really admirable the work done to build such a comprehensive model incorporating the wide array of processes involved in a soil-focused landscape evolution model. I have some comments below on specific things, but I want to acknowledge ahead of time that the model is pretty broad and these comments may not change the broader results.*

**Response:** Dear Harrison Gray,

Thank you for the nice remarks about the manuscript and the constructive comments. Below we will address the comments one-by-one. Your comments are marked in italic.

With best regards, on behalf of all authors,

Marijn van der Meij

***Main Comments:***

*Soil transport in ChronoLorica*

*One thing that I wondered about is whether the treatment of soil horizontal and vertical transport is internally-consistent. An example may be the comparison between the soil creep function and the bioturbation function along with the particle transport formulae of Anderson (2015) and Furbish et al. (2018b). There isn't enough detail in this section for me to fully understand how the model is working, but I wonder are the ddCR and ddBT values consistent with that input into Anderson/Furbish? If so, how is this done?? The second point on this is that it isn't clear what the justification is for exponential decay functions for soil creep and bioturbation. It would be helpful to back this up with references. Perhaps with some of the Young's pit studies?*

**Response:** In our model, vertical transport is driven by bioturbation, while horizontal transport is driven by soil creep. These processes influence the bulk of the soil, by mixing and transporting material between different layers. Our depth parameters $dd_{CR}$ and $dd_{BT}$, with the unit m$^{-1}$, determine how the intensity of these processes changes with soil depth. This is similar to the exponential speed profile reported in Anderson (2015). Anderson (2015) divides the soil depth by his depth scale of creep ($z_a = 0.15$ m), while we multiply our soil depth with $dd_{CR}$ and $dd_{BT}$. The inverse of $z_a$ is $1 / 0.15$ m $= 6.67$ m$^{-1}$, which is consistent with our depth parameters of 5 m$^{-1}$. We will better support our parameter selection in the manuscript, including this example.

We chose to work with an exponential decay function for creep and bioturbation, because this is the standard in the Lorica model and its predecessors (Vanwalleghem et al., 2013; Temme and Vanwalleghem, 2016). This shape is found in many soil processes and properties (Minasny et al., 2016) and represents diminishing temperature and soil moisture variations with depth (Amenu et al., 2005), biological activity for some organisms (Canti, 2003), as well as root distributions of some plants (Gregory, 2006). However, we're aware that the

exponential profile is not valid for all settings, as the references above also state. We will mention the selection of the exponential profile, and its alternatives, in the manuscript. We will also emphasize that for in each field setting there are different organisms and processes responsible for soil mixing and transport. For each study, the processes, depth functions and parameters should thus be derived from field data or similar studies.

*As another comment on this: On line 119, the authors say they use the formulae of Anderson (2015) and Furbish et al. (2018b) to model downhill transport of soil particles. However, these papers disagree with each other on the base principles of how soil moves downhill with the Anderson study assuming a continuum-style flux of soil (basically soil treated as a fluid) and the Furbish et al. study explicitly treating the soil particles as non-fluid with the advection-diffusion style equations describing the ensemble averaged conditions of soil particle transport using statistical mechanics. The advection/diffusion equations describe the flux of probability of the expected value of the ensemble average. The particle transport was handled with different random-walk equations. This in effect means that if you use Furbish theory, you cannot calibrate your OSL/cosmo field data against the advection/diffusion model because the model and data are two fundamentally different things. The model being a theoretical average of a uncountable number of theoretical soils. In contrast if you use the Anderson approach, you will be wrong because soil doesn't follow continuum mechanics as in the base assumptions of that paper.*

**Response:** Thank you for the clarification. We actually borrow from both descriptions, but we approach the issue from the bulk soil instead of from the individual particles. The bulk soil moves downslope, following diffusive transport (faster on steeper slopes). The changes in downslope velocity with soil depth is similar to the exponential profile of Anderson (2015) and Figure 6 of Gray et al. (2019), that refers to the Furbish theory. Vertical transport of soil particles is governed by the bioturbation process, that follows the same exponential depth dependence, meaning that there is more material exchange in soil layers nearer to the surface.

In ChronoLorica, the horizontal and vertical transport of bulk soil is coupled to the stochastic transport of individual particles. The fractions of bulk soil that is transported from a layer is used to randomly assess whether a particle is transported as well. Each particle follows its own unique trajectory.

The distribution of particle ages in certain soil layer can be expressed as a probability density functions (PDFs) of particle ages. These PDFs can be compared with PDFs of measured OSL particle ages. Calibration or validation can be done by comparing the PDFs, or specific statistics calculated from the PDFs, such as mean, median or spread.

We will rephrase the sentence on line 119, to clarify how we model soil and consequently particle transport.

*OSL physics*

*One thing that this paper made me think was that it would potentially be useful for the authors to directly simulate the luminescence. It seems like the authors get into a high level of detail with the cosmogenic physics but not the OSL physics. I think this is worth exploring in the model because the model is intended to be an explicit coupled soil-landscape-geochronometers model, yet the physics of cosmogenic is treated very in depth but the OSL isn't. Right now the model feels very focused on the cosmo*

**Response:** We simplified physics behind OSL and cosmogenic nuclides to match the reduced complexity of our model. In case of particle ages, this simplification leads to the tracing of individual particle ages, rather than the dose rate and palaeodose of each particle. This approach is sufficient to simulate the required age distributions at this stage. In the case of cosmogenic nuclides, the simulated processes are indeed a bit more detailed, because the nuclides have different production pathways, the production is linked to soil depth and certain particle sizes and behave differently in soils. Nonetheless, also these physics are simplified compared to conventional cosmogenic nuclide models (Balco, 2017).

*One thing that brought this up is the assumption that the burial age equals the OSL age, which I think isn't always an easy assumption in soils! As a particle travels through various soil layers, the background dose rate, DR(z), can change due to a variety of processes, but particularly with soil density and water content which affects the density of the natural background radiation intensity and the cosmogenic radiation flux. I didn't see any content or discussion on how this could affect the OSL geochronometer results, but I could imagine that the burial age and the OSL age could vary a lot (it does from my field experience with OSL in soil). I think it is important for the authors to explore this assumption and show, perhaps with a sensitivity analysis that it does or*

*does not matter.*

*As an admission, I assumed constant DR with soil depth in past work and I think Furbish et al. (2018b) made a case for why this does not matter but I don't remember fully. I'm bringing this up because treatment of specific soil layers seems to be an important benefit of Chronolorica.*

*One could model the change in luminescence (and assumed an idealized luminescence geochronometer, the luminescence grows in at $dL(z)/dt = DR(z)/D0 * (Ls – L(z))$ where L is luminescence, z is vertical height in the soil, t is time, DR(z) is dose rate, D0 is a dose e-folding scale, and Ls is the luminescence at saturation. A really good source for this type of modeling is Brown, N. D. (2020). Which geomorphic processes can be informed by luminescence measurements?. Geomorphology, 367, 107296 Where the author gives the equations that could be directly incorporated into the model.*

**Response:** The reviewer points out an important shortcoming in the way OSL ages are currently calculated for bioturbated soils. In most experimental studies it is assumed that the dose rate at sampling position is the best estimate for all particles at this position, which only is a fair assumption if a constant dose rate with soil depth is assumed in the case of bioturbation studies, where mobile particles are studied. Therefore, we think that our approach of tracing particle ages is sufficient for comparison with most measured ages.

However, we agree that spatial (and temporal) changes in the effective dose rate that a particle receives during its soil passage can have a large effect on the age determinations, and that ChronoLorica is a suitable tool to explore the impact of spatiotemporal variations of this dose rate on particle OSL ages. This can be interesting for future work for sure. We therefore will add this as a possible application of the model in Section 5.3.

***Minor Comments:***

*Line 41: if helpful, Gray et al 2019 has a section on soil mixing and methods that might be helpful for this review paragraph: Gray, H. J., Jain, M., Sawakuchi, A. O., Mahan, S. A., & Tucker, G. E. (2019). Luminescence as a sediment tracer and provenance tool. Reviews of Geophysics, 57(3), 987-1017.*

**Response:** This paper is indeed a useful addition to the review paragraph. We will add the reference to the list at the end of this paragraph.

*Line 64: Hmm. The comparison with field studies is a bit weird as SLEMs are hypothetical scenarios but are not actually reality in the way that field studies are.*

**Response:** We understand your concerns with this statement, but we prefer to keep this comparison in the manuscript to show the differences in scale and continuity between field and model data. We will nuance this statement by adding a sentence the end of this paragraph to indicate that the model data are hypothetical and simplified compared to the field data.

*Line 198, 459: 10 mm seems high for light penetration in soil. See: Ciani, A., Goss, K. U., & Schwarzenbach, R. P. (2005). Light penetration in soil and particulate minerals. European journal of soil science, 56(5), 561-574.*

**Response:** In the paper you cite, the authors create samples of particles smaller than 0.355 mm, which they press into a container with a sheet of paper on top to provide some surface roughness. These are then measured for reflectance. We don't think that this method is representative for light penetration in soil in the context of OSL dating, because the sand fraction is removed, and effects of soil structure and soil surface roughness are left out. We think these factors all lead to higher porosity and deeper light penetration. Moreover, the measurement time is probably much shorter than the annual time step in our model. Sub-annual mixing can transport bleached particles deeper in the soil as well.

In our simulations, we used a light penetration depth of 5 mm (Table 1), which in between the 1 and 10 mm used in Furbish et al. (2018), and which is consistent with long-term bleaching depths in soils (Sellwood et al., 2019). We expect that the light penetration depth in soils is in this order of magnitude, but experimental data is required to make better supported estimates, as we also address in lines 457-462.

**References**

Amenu, G. G., Kumar, P., and Liang, X.-Z.: Interannual variability of deep-layer hydrologic memory and mechanisms of its influence on surface energy fluxes, Journal of climate, 18, 5024–5045, 2005.

Anderson, R. S.: Particle trajectories on hillslopes: Implications for particle age and 10Be structure, Journal of Geophysical Research: Earth Surface, 120, 1626–1644, https://doi.org/10.1002/2015JF003479, 2015.

Balco, G.: Production rate calculations for cosmic-ray-muon-produced 10Be and 26Al benchmarked against geological calibration data, Quaternary Geochronology, 39, 150–173, https://doi.org/10.1016/j.quageo.2017.02.001, 2017.

Canti, M. G.: Earthworm Activity and Archaeological Stratigraphy: A Review of Products and Processes, Journal of Archaeological Science, 30, 135–148, https://doi.org/10.1006/jasc.2001.0770, 2003.

Furbish, D. J., Roering, J. J., Keen-Zebert, A., Almond, P., Doane, T. H., and Schumer, R.: Soil particle transport and mixing near a hillslope crest: 2. Cosmogenic nuclide and optically stimulated luminescence tracers, Journal of Geophysical Research: Earth Surface, 123, 1078–1093, https://doi.org/10.1029/2017JF004316, 2018.

Gray, H. J., Jain, M., Sawakuchi, A. O., Mahan, S. A., and Tucker, G. E.: Luminescence as a sediment tracer and provenance tool, Reviews of Geophysics, 57, 987–1017, 2019.

Gregory, P. j.: Roots, rhizosphere and soil: the route to a better understanding of soil science?, European Journal of Soil Science, 57, 2–12, https://doi.org/10.1111/j.1365-2389.2005.00778.x, 2006.

Minasny, B., Stockmann, U., Hartemink, A. E., and McBratney, A. B.: Measuring and Modelling Soil Depth Functions, in: Digital Soil Morphometrics, edited by: Hartemink, A. E. and Minasny, B., Springer International Publishing, Cham, 225–240, https://doi.org/10.1007/978-3-319-28295-4_14, 2016.

Sellwood, E. L., Guralnik, B., Kook, M., Prasad, A. K., Sohbati, R., Hippe, K., Wallinga, J., and Jain, M.: Optical bleaching front in bedrock revealed by spatially-resolved infrared photoluminescence, Sci Rep, 9, 2611, https://doi.org/10.1038/s41598-019-38815-0, 2019.

Temme, A. J. A. M. and Vanwalleghem, T.: LORICA – A new model for linking landscape and soil profile evolution: development and sensitivity analysis, Computers & Geosciences, 90, 131–143, https://doi.org/10.1016/j.cageo.2015.08.004, 2016.

Vanwalleghem, T., Stockmann, U., Minasny, B., and McBratney, A. B.: A quantitative model for integrating landscape evolution and soil formation, J Geophys Res-Earth, 118, 331–347, https://doi.org/10.1029/2011jf002296, 2013.